# Synaptic transmission parallels neuromodulation in a central food-intake circuit

Philipp Schlegel[1], Michael J Texada[2], Anton Miroschnikow[1], Andreas Schoofs[1], Sebastian Hückesfeld[1], Marc Peters[1], Casey M Schneider-Mizell[2], Haluk Lacin[2], Feng Li[2], Richard D Fetter[2], James W Truman[2], Albert Cardona[2], Michael J Pankratz[1]*

[1]Department of Molecular Brain Physiology and Behavior, LIMES Institute, University of Bonn, Bonn, Germany; [2]Janelia Research Campus, Howard Hughes Medical Institute, Ashburn, United States

**Abstract** NeuromedinU is a potent regulator of food intake and activity in mammals. In *Drosophila*, neurons producing the homologous neuropeptide hugin regulate feeding and locomotion in a similar manner. Here, we use EM-based reconstruction to generate the entire connectome of hugin-producing neurons in the *Drosophila* larval CNS. We demonstrate that hugin neurons use synaptic transmission in addition to peptidergic neuromodulation and identify acetylcholine as a key transmitter. Hugin neuropeptide and acetylcholine are both necessary for the regulatory effect on feeding. We further show that subtypes of hugin neurons connect chemosensory to endocrine system by combinations of synaptic and peptide-receptor connections. Targets include endocrine neurons producing DH44, a CRH-like peptide, and insulin-like peptides. Homologs of these peptides are likewise downstream of neuromedinU, revealing striking parallels in flies and mammals. We propose that hugin neurons are part of an ancient physiological control system that has been conserved at functional and molecular level.

*For correspondence: pankratz@uni-bonn.de

**Competing interests:** The authors declare that no competing interests exist.

## Introduction

Multiple studies have demonstrated functional conservation of fundamental hormonal systems for metabolic regulation in mammals and *Drosophila*. This includes insulin (*Ikeya et al., 2002*; *Rulifson et al., 2002*), glucagon (*Kim and Rulifson, 2004*), and leptin (*Rajan and Perrimon, 2012*). In addition to these predominantly peripherally released peptides, there is a range of neuropeptides that are employed within the central nervous systems (CNS) of vertebrates and have homologs in invertebrates, e.g. neuropeptide Y (NPY), corticotropin-releasing hormone (CRH) or oxytocin/vaso-pressin (*Nässel and Winther, 2010*; *Nässel and Wegener, 2011*; *Grimmelikhuijzen and Hauser, 2012*; *Mirabeau and Joly, 2013*; *Jékely, 2013*).

Among these, neuromedinU (NMU) is known for its profound effects on feeding behavior and activity; NMU inhibits feeding behavior (*Howard et al., 2000*), promotes physical activity (*Novak et al., 2007*; *Chiu et al., 2016*), and is involved in energy homeostasis (*Nakazato et al., 2000*; *Ivanov et al., 2002*) and stress response (*Hanada et al., 2001*; *Zeng et al., 2006*). Hugin is a member of the pyrokinin/PBAN (pheromone biosynthesis activating neuropeptide) peptide family and a *Drosophila* homolog of NMU that has recently gained traction due to similar effects on behavior in the fly: increased hugin signaling inhibits food intake and promotes locomotion (*Melcher et al., 2006*; *Schoofs et al., 2014*; *Bader et al., 2007b*). In mammals, distribution of the NMU peptide, NMU-expressing cells and NMU-positive fibers is wide and complex. High levels of

NMU have been reported in the arcuate nucleus of the hypothalamus, the pituitary, the medulla oblongata of the brain stem, and the spinal cord (*Domin et al., 1987*; *Ballesta et al., 1988*; *Howard et al., 2000*; *Ivanov et al., 2004*). The number of neurons involved and their morphology is unknown. In *Drosophila*, the distribution of hugin is less complex, yet similar: the peptide is produced by neurons in the subesophageal zone that have hugin-positive projections into the ring gland, the pars intercerebralis and ventral nerve cord (*Melcher and Pankratz 2005*) (*Figure 1*). While comparisons across large evolutionary distances are generally difficult, these regions of the fly brain were suggested to correspond to aforementioned regions of NMU occurrence based on morphological, genetic and functional similarities (*Ghysen, 2003*; *Hartenstein, 2006*). Consequently, NMU/hugin has previously been referred to as a clear example of evolutionary constancy of peptide function (*Taghert and Nitabach, 2012*).

Although functional and morphological aspects of neurons employing either neuropeptide have been extensively studied in the past, knowledge about their connectivity is fragmentary. While large-scale connectomic analyses in vertebrates remain challenging, generation of high-resolution connectomes has recently become feasible in *Drosophila* (*Ohyama et al., 2015*; *Berck et al., 2016*; *Fushiki et al., 2016*; *Schneider-Mizell et al., 2016*). We took advantage of this and performed an integrated analysis of synaptic and G-protein-coupled receptor (GPCR)-mediated connectivity of hugin neurons in the CNS of *Drosophila*. Our data demonstrates that hugin neurons employ small molecule transmitters in addition to the neuropeptide. We identify acetylcholine as a transmitter that is employed by hugin neurons and find that it is required for their effect on feeding behavior. Next, we show that hugin neurons form distinct units, and demonstrate that clusters of neurons employing the same neuropeptide are remarkably different in their synaptic connectivity. One unit of hugin neurons is presynaptic to subsets of median neurosecretory cells (mNSCs) in the protocerebrum. In parallel to the synaptic connectivity, mNSCs also express the G-protein-coupled receptor PK2-R1, a hugin receptor, rendering them likely targets of both fast synaptic transmission and neuromodulatory effects from hugin neurons. These mNSCs produce diuretic hormone 44 (DH44, a CRH-like peptide) and *Drosophila* insulin-like peptides, both of which have mammalian homologs that are likewise downstream of NMU (*Wren et al., 2002*; *Malendowicz et al., 2012*). Endocrine function is essential to ensure homeostasis of the organism and coordinate fundamental behaviors, such as feeding, mating and reproduction, and acts as integrator of external and internal sensory cues (*Swanson, 2000*). Consequently, connections between sensory and endocrine systems are found across species (*Yoon et al., 2005*; *Tessmar-Raible et al., 2007*; *Strausfeld 2012*; *Abitua et al., 2015*). We show that hugin neurons receive chemosensory input in the subesophageal zone (SEZ), thereby linking chemosensory and neuroendocrine systems.

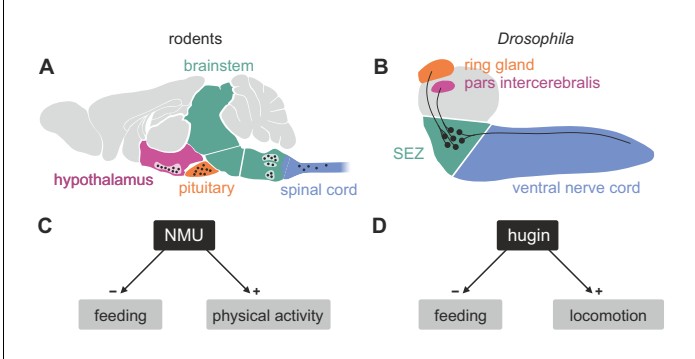

**Figure 1.** Comparison of mammalian neuromedinU and *Drosophila* hugin. (**A**) NeuromedinU (NMU) is widely distributed in the rodent CNS. NMU peptide, NMU-expressing cells and NMU-positive fibers are found in several regions of the brain stem, hypothalamus, pituitary and spinal cord (black dots). The number of neurons and their morphology is unknown. (**B**) In *Drosophila*, distribution of the homologous neuropeptide hugin is less complex and well known: hugin is expressed by sets of neurons in the subesophageal zone (SEZ) that project into the pars intercerebralis, ring gland and ventral nerve cord. (**C**, **D**) Increased NMU and hugin signaling has similar effects: feeding behavior is decreased, whereas physical activity/locomotion is increased.

## Results

### Input and output compartments of hugin neurons

The *hugin* gene is expressed in only 20 neurons in the *Drosophila* CNS. This population comprises interneurons, which are confined within the CNS, as well as efferent neurons, which leave the CNS. The interneuron type can be subdivided into those projecting to the protocerebrum (hugin-PC, eight neurons) or the ventral nerve cord (hugin-VNC, four neurons). The efferent type can be subdivided into those projecting to the ring gland (hugin-RG, four neurons) or the pharynx (hugin-PH, four neurons) (*Figure 2A*) (*Bader et al., 2007a*). Based on these morphological features, we first reconstructed all hugin neurons in an ssTEM volume covering an entire larval CNS and the major neuroendocrine organ, the ring gland (*Figure 2B*; see Materials and methods for details). We then localized synaptic sites, which could be readily identified as optically dense structures (*Prokop and Meinertzhagen, 2006*). Comparing neurons of the same class, we found the number as well as the distribution of pre- and postsynaptic sites to be very similar among hugin neurons of the same class (*Figure 2C–E*, *Video 1*). Presynaptic sites are generally defined as having small clear core vesicles (SCVs) containing classic small molecule transmitter for fast synaptic transmission close to the active zone (*Prokop and Meinertzhagen, 2006*). Efferent hugin neurons (hugin-RG and hugin-PH) showed essentially no presynaptic sites (<1 average/neuron) within the CNS and we did not observe any SCVs. For hugin-RG neurons, membrane specializations resembling presynaptic sites were evident at their projection target, the ring gland. These sites did contain close-by DCVs but no SCVs and had in many cases no corresponding postsynaptic sites in adjacent neurons. Instead they bordered haemal space indicating neuroendocrine release (*Figure 2—figure supplement 1A*). The configuration of hugin-PH terminals is unknown as their peripheral target was outside of the ssTEM volume. For the interneuron classes (hugin-PC and hugin-VNC), we found SCVs at larger presynaptic sites, indicating that they employ classic neurotransmitter in addition to the hugin peptide (*Figure 2—figure supplement 1B,C*). Hugin-PC and hugin-VNC neurons' projections represent mixed synaptic input-output compartments as they both showed pre- as well as postsynaptic sites along their neurites (*Figure 2D,E*).

All hugin neurons receive inputs within the SEZ [previously called subesophageal ganglion (SOG)], a chemosensory center that also houses the basic neuronal circuits generating feeding behavior (*Hückesfeld et al., 2015*). However, only the hugin-PC neurons showed considerable numbers of synaptic outputs in the SEZ, consistent with their previously reported effects on feeding (*Schoofs et al., 2014*; *Hückesfeld et al., 2016*) (*Figure 2E*).

### Acetylcholine is a co-transmitter in hugin neurons

The existence of presynaptic sites containing SCVs in addition to large DCVs led to the assumption that hugin-PC and hugin-VNC (possibly also hugin-PH neurons) employ small molecule neurotransmitters parallel to the hugin neuropeptide. To address this, we checked for one of the most abundantly expressed neurotransmitter in the *Drosophila* nervous system: acetylcholine (ACh) (*Yasuyama and Salvaterra, 1999*; *Salvaterra and Kitamoto, 2001*). In the past, immunohistochemical and promoter expression analyses of choline acetyltransferase (ChAT), the biosynthetic enzyme for ACh, were successfully used to demonstrate cholinergic transmission (*Barnstedt et al., 2016*; *Miyamoto, 2012*; *Yapici et al., 2016*). We used both, anti-ChAT antibody as well as a ChAT promoter analysis, and investigated co-localization with hugin neurons. In the EM data, we found hugin neurons to have comparatively few SCVs, suggesting only low amounts of small transmitters. In addition, ChAT is preferentially localized in the neuropil and less so in the somas (*Sámano et al., 2006*). Consistent with this, we found that ChAT immunoreactivity in hugin cell bodies was relatively low and varied strongly between samples. Therefore, we quantified the anti-ChAT signal to show that while ChAT levels were in some cases indiscernible from the background, overall highest levels of ChAT were found in hugin-PC and hugin-VNC/PH neurons (*Figure 3A*). Note that while hugin-PC and hugin-RG neurons were easily identifiable based on position and morphology, hugin-PH and hugin-VNC neurons usually clustered too tightly to be unambiguously discriminated and were thus treated as a single mixed group. Similar to the immunohistochemical analysis, the ChAT promoter (ChAT-GAL4) drove expression of a fluorescent reporter in all hugin-PC neurons plus a subset of

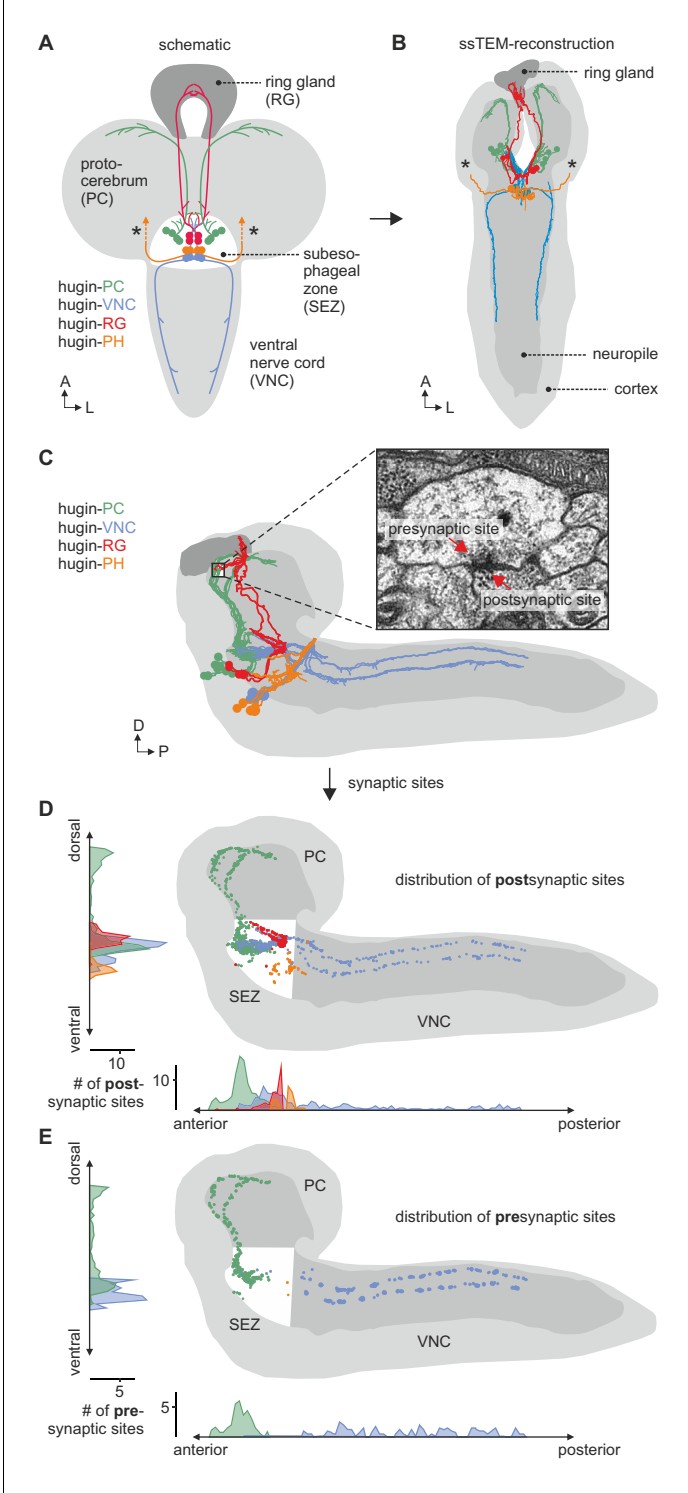

**Figure 2.** EM reconstruction of hugin neurons and their synaptic sites. (A) Hugin neurons are known to form four morphologically distinct classes: hugin-PC (protocerebrum), hugin-VNC (ventral nerve cord), hugin-RG (ring gland) and hugin-PH (pharynx, asterisks mark nerve exit sites). (B) Reconstruction of all hugin neurons based on serial section electron microscopy (EM) of an entire larval brain. (C–E) Spatial distribution of synaptic sites for all hugin classes. Hugin interneurons (hugin-PC and hugin-VNC) show mixed input/output compartments, and presynaptic sites indicate the existence of a small molecule transmitter in addition to the hugin neuropeptide. In contrast, Hugin-RG and hugin-PH show almost exclusively postsynaptic sites within the CNS. Each dot in D and E
*Figure 2 continued on next page*

*Figure 2 continued*

represents a single synaptic site. Graphs show distribution along dorsal-ventral and anterior-posterior axis of the CNS. See also **Video 1**.

The following figure supplement is available for figure 2:

**Figure supplement 1.** Exemplary synaptic sites in the ssTEM volume.

hugin-VNC/PH neurons (**Figure 3B**). Hugin-RG showed weak ChAT signal with either method, consistent with these neurons lacking SCVs in the EM data.

These findings suggested that ACh may be a co-transmitter in hugin neurons. We previously demonstrated that RNAi-induced knockdown of the hugin neuropeptide rescues the phenotype of feeding suppression caused by induced activation of hugin neurons in behavioral and electrophysiological experiments (**Schoofs et al., 2014**). Here, we present the knockdown of ChAT using an established UAS-ChAT-RNAi line (**Plaçais et al., 2013**; see Materials and methods). Under unimpeded conditions (i.e., without ChAT knockdown), activation of hugin neurons leads to severe decrease of food intake in intact larvae. This decrease in food intake was rescued by knockdown of ChAT in hugin neurons to a similar degree as the knockdown of the hugin neuropeptide itself (**Figure 3C**).

In addition to a general decrease in food intake, activation of hugin neurons leads to a decrease in cycle frequency of pharyngeal pump motor activity (**Schoofs et al., 2014**). We used extracellular recordings of the antennal nerve (AN) in isolated CNS for precise monitoring of motor patterns of the pharyngeal pump (**Schoofs et al., 2009**). As for the food intake, knockdown of ChAT in hugin neurons also rescued the suppressive effect of hugin neuron activation on pharyngeal pumping (**Figure 3D**). Taken together, these data clearly demonstrate that ACh plays a functional role in hugin neurons. Moreover, this suggests that hugin neuropeptide and ACh have to be employed together in order to regulate feeding behavior.

## Hugin classes form distinct units that share synaptic partners

Reconstruction of hugin neurons and localization of synaptic sites revealed that neurons of the two interneuron classes, hugin-PC and hugin-VNC, were reciprocally connected along their main neurites to ipsilateral neurons of the same class (**Figure 4**, **Figure 2—figure supplement 1E,F**). These connections made up a significant fraction of each neuron's total synaptic connections, implying that their activity might be coordinately regulated.

We therefore further explored the different classes within the population of hugin-producing neurons, asking whether hugin classes establish functional units or whether they are independently wired. To this end, we reconstructed 177 pre- and postsynaptic partners of hugin neurons (**Figure 5A**, see Materials and methods for details). First, we found that neurons of the same hugin class were connected to the same pre- and postsynaptic partners. Furthermore, most synaptic partners were connected exclusively to neurons of a single hugin class (**Figure 5B**). Second, pre- and postsynaptic partners of each hugin class resided in different parts of the CNS (**Figure 5C**; **Video 2**). For hugin-RG and hugin-PH, the vast majority of synapses were made with interneurons, 93 ± 4% and 97 ± 3%, respectively. This percentage was lower for hugin-PC (66 ± 6%) and hugin-VNC (81 ± 2%). To our knowledge, none of these interneuron partners

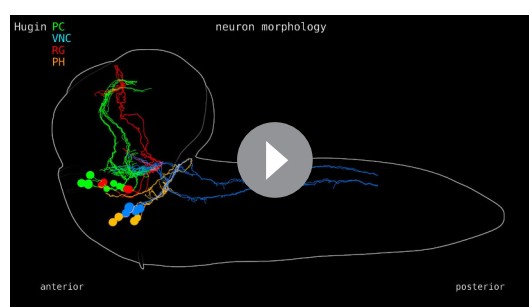

**Video 1.** Morphology of hugin-producing neurons. Video shows morphology of hugin-producing neurons as well as distribution of their presynaptic and postsynaptic sites. Hugin interneurons (PC and VNC) have mixed input-output compartments, whereas efferent hugin neurons (PH and RG) show almost exclusively postsynaptic sites within the CNS. Outlines of the CNS including the ring gland are shown in white.

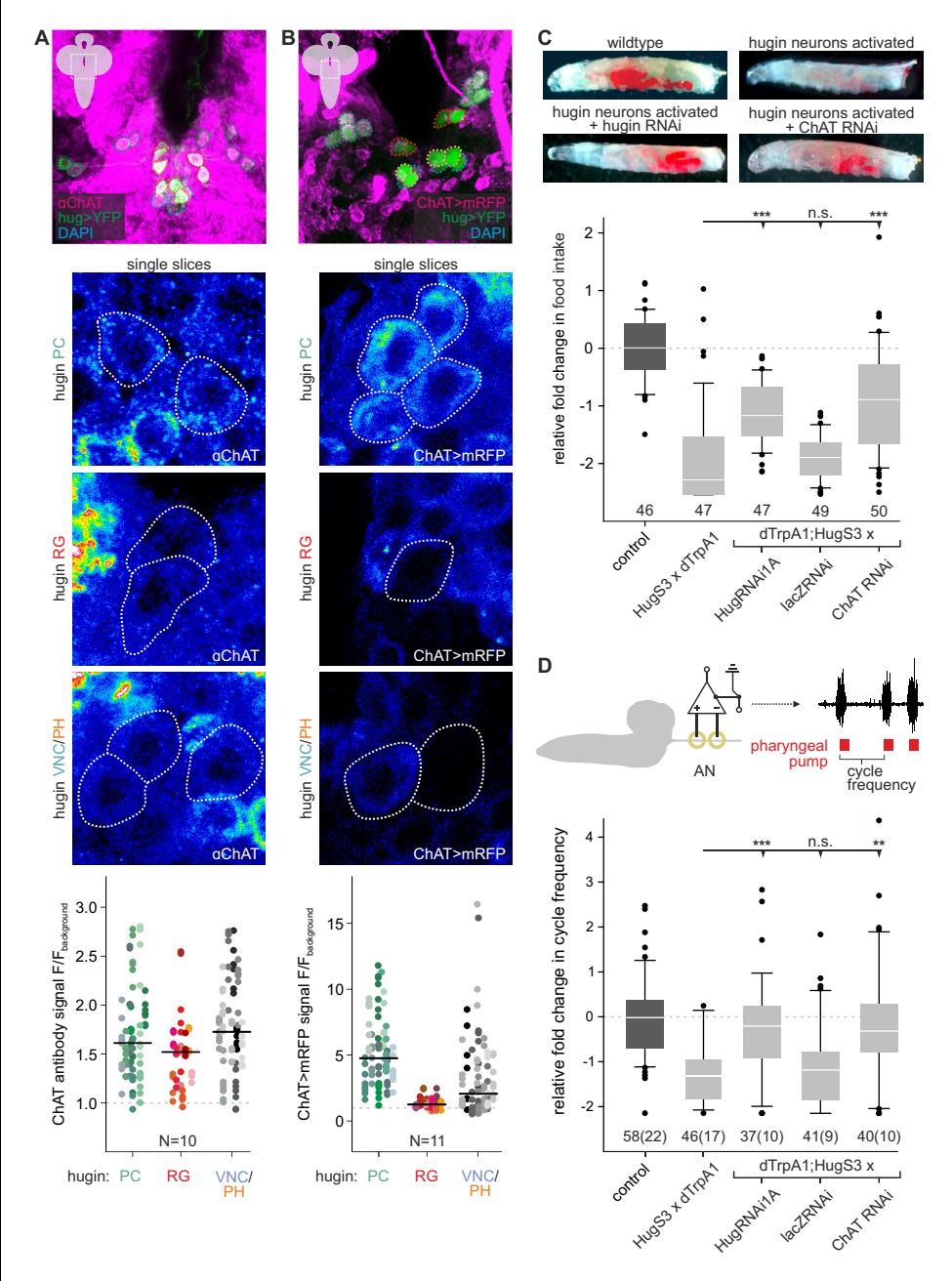

**Figure 3.** Acetylcholine (ACh) is a neurotransmitter of hugin neurons. (A,B) Co-localization of the biosynthetic enzyme for ACh, choline acetyltransferase (ChAT), in hugin neurons using a ChAT antibody (A) or a ChAT promoter-GAL4 driving a fluorescent reporter (B). ChAT immunoreactivity was variable but strongest signals were found in hugin-PC and hugin-VNC/PH neurons. Similarly, ChAT-GAL4 consistently drove expression in hugin-PC and subsets of hugin-VNC/PH. Shown are exemplary scans and quantification of ChAT co-localization in the different hugin classes. Note that while hugin-PC and hugin-RG neurons are easily identifiable, hugin-PH and hugin-VNC neurons were usually too close to be unambiguously discriminated and were thus treated as a single mixed group. Each data point in the dot plots represents a single hugin neuron. Horizontal lines mark median. (C, D) ACh is necessary for the effect of hugin neurons' activation on food intake (C) and pharyngeal pumping (D). Food intake was measured in intact larvae. Pharyngeal pumping was monitored by extracellular recordings of the antennal nerve (AN) and analyzed with respect to the cycle frequency of the motor patterns. Activation of hugin neurons using the thermosensitive cation channel dTrpA1 (HugS3-GAL4 x UAS-dTrpA1) led to a decrease in food intake and pharyngeal pump activity compared to the control (OrgR, OrgR x UAS-dTrpA1). Knockdown of either the hugin neuropeptide or ChAT but not LacZ control (UAS-dTrpA1;HugS3-GAL4 x UAS-RNAi) rescued the effect

*Figure 3 continued on next page*

*Figure 3 continued*

of hugin neuron activation on food intake as well as on pharyngeal pumping. For details see Materials and methods, and *Schoofs et al., (2014)*. Numbers below box plots give N [C: # larvae; D: # trials (# larvae)]. Mann-Whitney Rank Sum Test (*** = p<0.001; **=p < 0.01).

have been previously described, making it difficult to speculate on their functions at this point. Non-interneuron partners will be described in the following sections. In summary, these findings show that neurons of each hugin class form complex microcircuits that are largely separate from one another.

## Hugin neurons receive diverse chemosensory synaptic input

Hugin neurons have a significant number of their incoming synapses (63 ± 22%) within the SEZ. This region of the CNS is analogous to the brainstem and is a first-order chemosensory center that receives input from various sensory organs (*Ghysen, 2003*). In addition, subsets of hugin neurons were recently shown to be responsive to gustatory stimuli (*Hückesfeld et al., 2016*). We therefore searched for sensory inputs to hugin neurons and found a total of 68 afferent neurons that made synaptic contacts onto hugin neurons (*Figure 6A*). Two major groups emerged: a larger, morphologically heterogeneous group consisting of afferent neurons projecting through one of the pharyngeal nerves (the antennal nerve) and, unexpectedly, a second, more homogeneous group entering the CNS through abdominal (but not thoracic) nerves. We observed that the reconstructed afferent presynaptic partners of hugin neurons covered different parts of the SEZ. Thus, we sought to cluster these afferent neurons by computing the similarity in spatial distribution of their synaptic sites, termed synapse similarity score.

Clustering based on synapse similarity score resulted in seven different groups, each of them covering distinct parts of the SEZ (*Figure 6B*; *Video 3*; see Materials and methods for details). To address the issue of the origin of identified sensory inputs, we compared our data with previous descriptions of larval sensory neurons. It is well established that abdominal nerves innervate internal and external sensory organs of the peripheral nervous system. This includes proprioceptive (chordotonal), tactile, nociceptive (multi dendritic neurons) and a range of sensory neurons whose function is still unknown (*Hwang et al., 2007*; *Ghysen et al., 1986*;

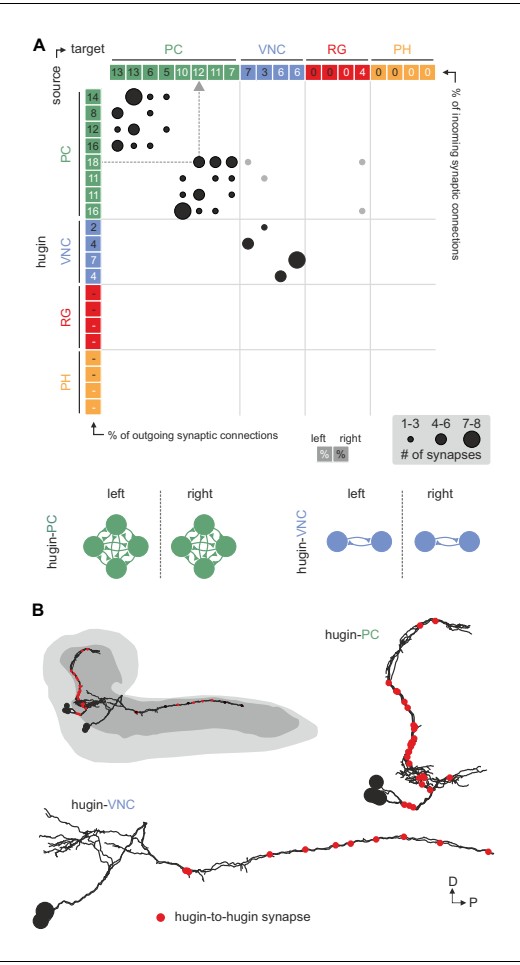

**Figure 4.** Hugin neurons synapse reciprocally within-class but not across-class. (**A**) Connectivity matrix of hugin to hugin connections. Each row indicates number of synaptic connections of given hugin neuron to other hugin neurons. Connections that could not be recapitulated for both hemisegments are grayed out. Numbers in colored boxes give % of incoming (x-axis) and outgoing (y-axis) synaptic connections of the respective hugin neuron. Hugin to hugin contacts are made between hugin interneurons of the same class, not between classes (see schematic). Note that efferent hugin neurons, hugin-RG and hugin-PH, do not have presynaptic sites. (**B**) Distribution of hugin-hugin synapses. Synaptic contacts between hugin-PC or hugin-VNC neurons are made along their main neurites. Only neurons of one hemisegment are shown.

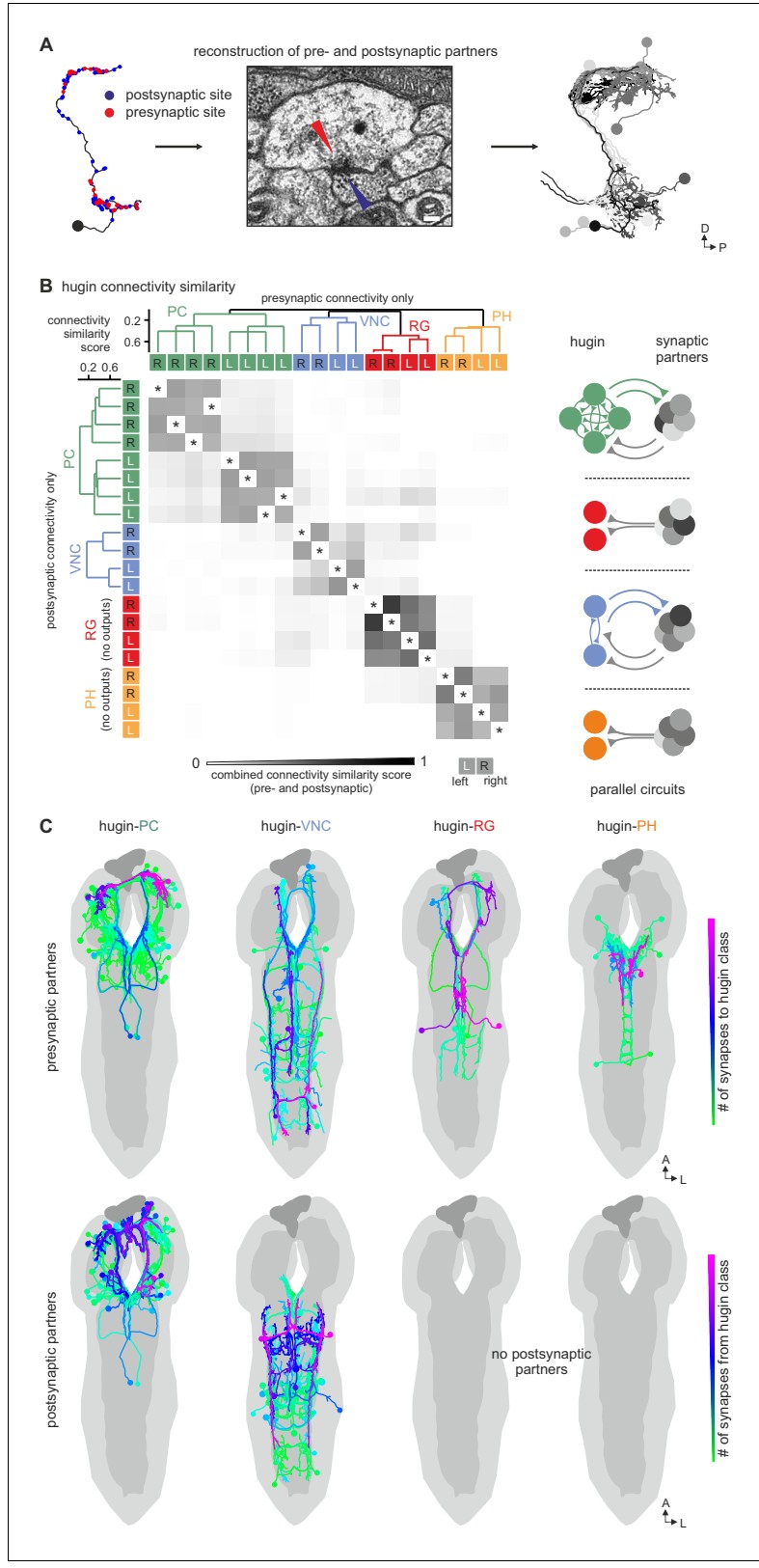

**Figure 5.** Each hugin class is part of a distinct microcircuit, weakly or not at all connected to those of the other classes. (**A**) Synaptic partners of hugin neurons were reconstructed. Pre- and postsynaptic partners of a single hugin-PC neuron are shown as example. (**B**) Comparison of hugin neurons' connectivity as measured by connectivity similarity score. High similarity score indicates a large fraction of shared synaptic partners connected

*Figure 5 continued on next page*

*Figure 5 continued*

by similar numbers of synapses. Neurons are ordered by dendrogram of similarity score of pre- (x-axis) and postsynaptic (y-axis) partners. Matrix shows combined pre- and postsynaptic similarity score. Self-self comparisons were omitted (asterisks). Hugin classes connect to unique sets of pre- and postsynaptic partners. Neurons of each hugin class have the same synaptic partners, and there is little to no overlap with other classes (see schematic). (C) Reconstructed pre- and postsynaptic partners by hugin class. Neurons are color-coded based on total number of synapses to given hugin class [minimum = 1; maximum (pre-/postsynaptic): hugin-PC = 53/16, hugin-VNC = 21/18, hugin-RG = 39/none, hugin-PH = 23/none]. Hugin-RG and hugin-PH neurons do not have postsynaptic partners within the CNS. See also *Video 2* and supplemental neuron atlas.

The following figure supplement is available for figure 5:

**Figure supplement 1.** Neurons connected by more than two synapses to hugin neurons were reliably reconstructed.

---

*Bodmer and Jan, 1987*). To our knowledge, no abdominal sensory neurons with projections into the SEZ such as the one observed presynaptic to hugin have been described. However, the majority of afferent neurons synapsing onto hugin neurons stems from the antennal nerve. This pharyngeal nerve carries the axons of gustatory receptor neurons (GRNs) from internal pharyngeal sensilla as well as those of olfactory receptor neurons (ORNs) and other GRNs from the external sensory organs (*Figure 6C,D*) (*Colomb et al., 2007*; *Vosshall and Stocker, 2007*). ORNs can be unambiguously identified as they target specific glomeruli of the antennal lobe (*Vosshall and Stocker, 2007*), but no such sensory neurons were found to directly input onto hugin neurons (*Figure 6—figure supplement 1*).

The GRNs likewise target restricted regions of the SEZ neuropil, but this is not as well characterized as the antennal lobes (*Colomb et al., 2007*; *Miyazaki and Ito, 2010*). The antennal nerve neurons that contact the hugin cells show the morphology of this large, heterogeneous population of GRNs (*Colomb, 2007*; *Kwon et al., 2011*). We thus compared our clustered groups with previously defined light microscopy-based gustatory compartments of the SEZ (*Colomb, 2007*). Groups 2 and 6, which cover the anterior-medial SEZ, likely correspond to two areas described as the target of GRNs from internal pharyngeal sensilla only (*Figure 6D*). The remaining groups were either not previously described or difficult to unambiguously align with known areas. Our division into groups is also reflected at the level of their connectivity to hugin neurons: sensory neurons of group 1 have synaptic connections to both hugin-PC and hugin-VNC neurons. Groups 2–5, encompassing the previously described pharyngeal sensilla, are almost exclusively connected to hugin-PC neurons. Group 6 sensory neurons make few synapses onto hugin-RG neurons. Group 7, encompassing the abdominal afferent neurons, is primarily presynaptic to hugin-VNC (*Figure 6E*).

The efferent type hugin neurons, hugin-PH and hugin-RG, show little to no sensory input. In contrast, the interneuron type hugin neurons, hugin-PC and hugin-VNC, receive a significant fraction of their individual incoming synaptic connections (up to 39%) from sensory neurons. Summarizing, we found two out of four types of hugin neurons to receive synaptic input from a large heterogeneous but separable population of sensory neurons, many of which are GRNs from external and internal sensory organs. Hugin-PC neurons were recently shown to be activated by bitter gustatory stimuli but not salt, fructose or yeast (*Hückesfeld et al., 2016*). Our data strongly suggests that this activation is at least partially based on monosynaptic

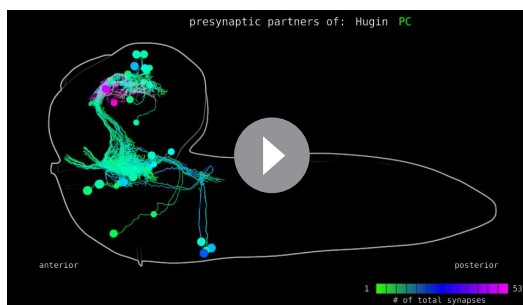

**Video 2.** Each class of hugin neurons connects to unique sets of synaptic partners. Video shows all reconstructed presynaptic and postsynaptic partners of hugin neurons (see *Figure 5C*). Neurons are colored by total number of synapses to/from given hugin class. Each hugin class forms distinct microcircuits with little to no overlap with those of other classes.

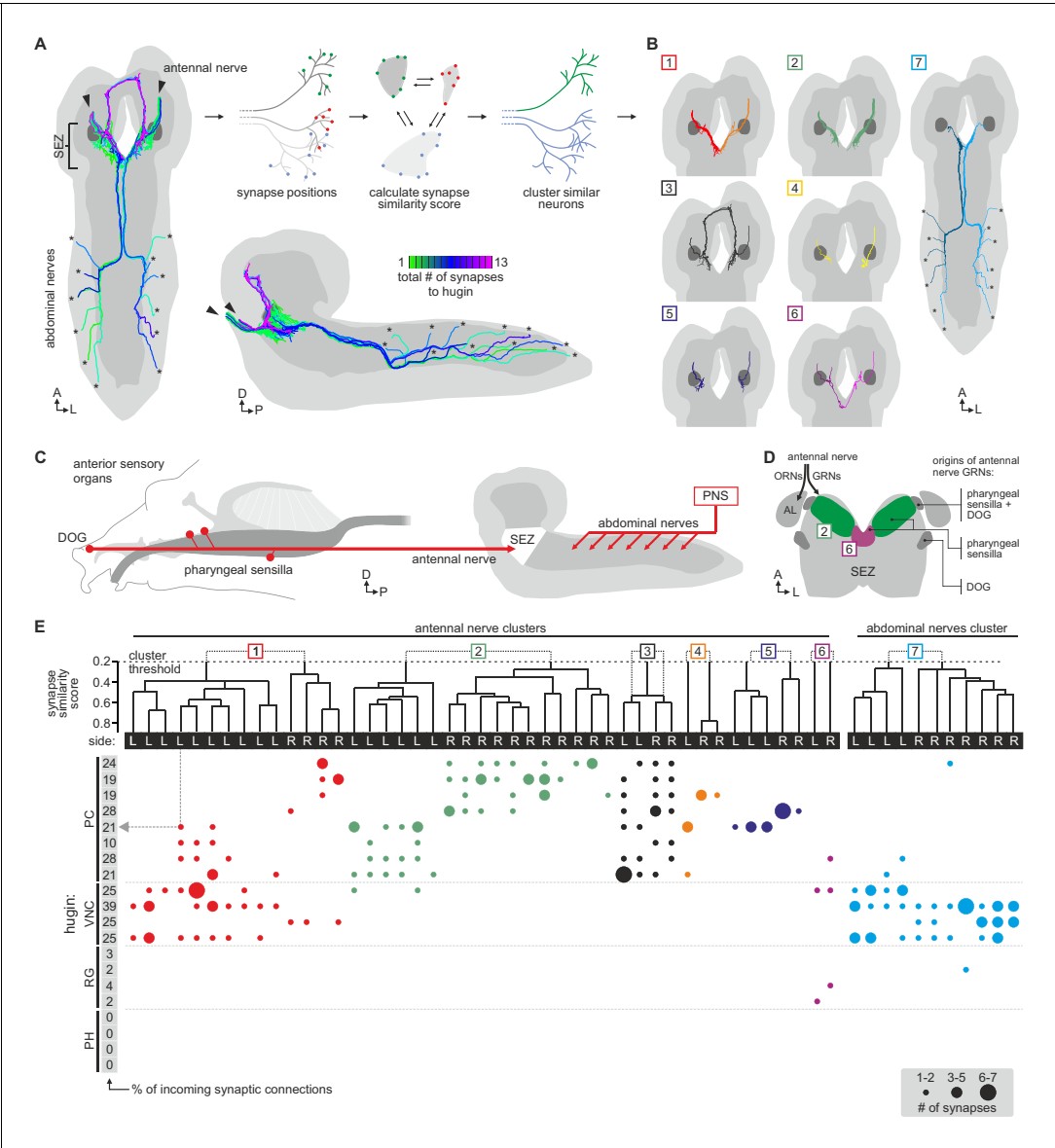

**Figure 6.** Each class of hugin neurons receives inputs from distinct subsets of sensory neurons. (A) Sensory inputs to hugin neurons enter the CNS via the antennal nerve (arrowheads) and abdominal nerves (asterisks). Neurons are color-coded based on total number of synapses to hugin neurons. (B) Sensory neurons clustered based on synapse similarity score. This score is computed from the spatial overlap of synaptic sites between two neurons. See also *Video 3*. (C) Potential origins of sensory inputs onto hugin neurons. The antennal nerve collects sensory axons from the dorsal organ ganglion (DOG) and pharyngeal sensilla. Abdominal nerves carry afferents from abdominal segments of the peripheral nervous system (PNS). (D) Target areas of antennal nerve chemosensory organs in the subesophageal zone (SEZ). Olfactory receptor neurons (ORNs) terminate in the antennal lobes (AL). Gustatory receptor neurons (GRNs) from different sensory organs cover distinct parts of the SEZ (based on (*Colomb et al., 2007*). (E) Connectivity matrix of sensory neurons onto hugin. Sensory neurons are ordered by dendrogram of synapse similarity score and rearranged to pair corresponding cluster of left (L) and right (R) hemisegment. Each row of the matrix shows the number of synaptic connections onto a single hugin neuron. Numbers in gray boxes along y-axis give percentage of synaptic input onto each hugin neuron represented as one neuron per row. A threshold of two synapse minimum was applied. See text for further details.

The following figure supplement is available for figure 6:

**Figure supplement 1.** Clustered synapses of sensory inputs to hugin neurons cover discrete parts of the subesophageal zone.

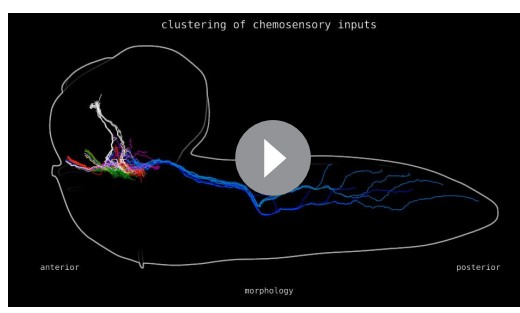

**Video 3.** Clusters of chemosensory neurons cover distinct areas of the subesophageal zone (SEZ). Video shows morphology and presynaptic sites of sensory inputs to hugin neurons. Neurons are clustered based on a synapse similarity score (see *Figure 6*). Each sphere represents a presynaptic site. Sphere size increases with the number of postsynaptically connected neuronal profiles for that synapse.

connections to GRNs. Moreover, the heterogeneity among the population of sensory neurons suggests that hugin-PC neurons do not merely function as simple relay station but rather fulfill an integrative function, for example between multiple yet-to-be-identified modalities or various external and internal sensory organs.

## Dual synaptic and peptide-receptor connection to the neuroendocrine system

NMU has been well studied in the context of its effect on the hypothalamo-pituitary axis. We therefore looked for similar motifs among the downstream targets of hugin neurons. The cluster of hugin-PC neurons projects their neurites from the SEZ to the protocerebrum, terminating around the pars intercerebralis. Median neurosecretory cells (mNSCs) in this area constitute the major neuroendocrine center in the CNS, homologous to the mammalian hypothalamus, and target the neuroendocrine organ of *Drosophila*, the ring gland (*Hartenstein, 2006*).

Three different types of mNSCs produce distinct neuropeptides in a non-overlapping manner: 3 mNSCs produce diuretic hormone 44 (DH44), 2 mNSCs produce Dromyosuppressin (DMS) and 7 mNSCs produce *Drosophila* insulin-like peptides (Dilps, thus called insulin-producing cells [IPCs]) (*Figure 7A*) (*Park et al., 2008*). We found that hugin-PC neurons make extensive synaptic contacts onto most but not all of the mNSCs (*Figure 7B*; *Figure 2—figure supplement 1G,H*). mNSCs of the pars intercerebralis derive from the same neuroectodermal placodes and develop through symmetric cell division (*de Velasco et al., 2007*). Among the mNSCs, IPCs have been best studied: they have ipsilateral descending arborizations into the SEZ and project contralaterally into the ring gland (*Rulifson et al., 2002*). In contrast, morphology of DH44- or DMS-producing mNSCs has been described in less detail. Our reconstruction showed that all reconstructed mNSCs have the exact same features, rendering them morphologically indistinguishable (*Figure 7C*). To assign identities to the reconstructed mNSCs, we hypothesized that similar to hugin neurons, the three types of mNSCs would differ in their choice of synaptic partners. We therefore reconstructed all presynaptic partners and calculated the connectivity similarity score between the mNSCs. Clustering with this similarity in connectivity resulted in three groups comprising 3, 2, and 7 neurons, coinciding with the number of neurons of the known types of mNSCs. We thus suggest that the group of three represents DH44-producing cells, the group of two represents DMS-producing cells and the group of seven represents the IPCs (*Figure 7D*).

On this basis, hugin-PC neurons make extensive synaptic contacts to the IPCs but less so to DMS- and DH44-producing mNSCs. In accordance with hugin-PC neurons using ACh as neurotransmitter, IPCs were previously shown to express a muscarinic ACh receptor (*Cao et al., 2014*). Whether additional ACh receptors are expressed is unknown. Overall, synapses between hugin-PC neurons onto mNSCs constitute a large fraction of their respective synaptic connections (hugin-PC: up to 35%; mNSCs: up to 17%). In support of a tight interconnection between hugin neurons and these neuroendocrine neurons, we found that most of hugin-PC neurons' presynaptic partners are also presynaptic to mNSCs (*Figure 7E*). These findings demonstrate that the neuroendocrine system is a major target of hugin neurons.

Unlike the small molecule messengers used for fast synaptic transmission, neuropeptides – such as hugin – are thought to be released independent of synaptic membrane specializations and are able to diffuse a considerable distance before binding their respective receptors. However, it has been proposed that neuropeptides released from most neurons act locally on cells that are either synaptically connected or immediately adjacent (*van den Pol 2012*). We therefore asked whether the synaptic connections between hugin-PC neurons and mNSCs would have a matching peptide-

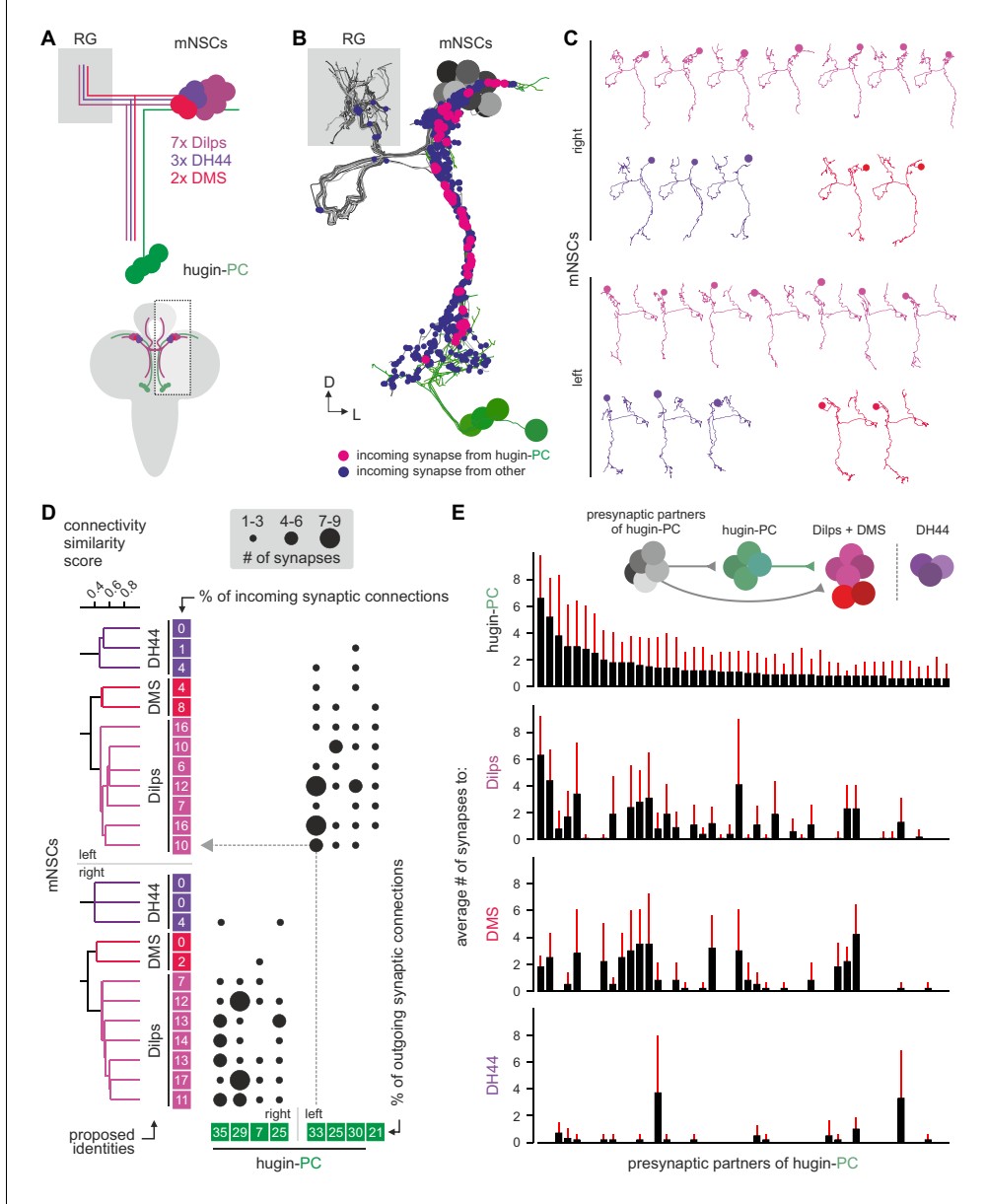

**Figure 7.** Hugin-PC neurons are presynaptic to all insulin-producing neurosecretory neurons. (A) Schematic of median neurosecretory cells (mNSCs) of the pars intercerebralis. mNSCs produce *Drosophila* insulin-like peptides (Dilps), diuretic hormone 44 (DH44) and Dromyosuppressin (DMS) in a non-overlapping manner. (B) EM reconstruction of all mNSCs and their synaptic contacts with hugin-PC neurons. (C) Ipsilateral mNSCs present similar arborizations, making morphological identification impossible. Instead mNSCs were categorized by connectivity (see D). (D) Synaptic partners of mNSCs were reconstructed and mNSCs were clustered based on connectivity similarity. This revealed three clusters consistently across both hemispheres that matched groups of 3 DH44-, 2 DMS- and 7 Dilps-producing cells (see text for details). Connectivity matrix shows that hugin-PC neurons primarily target Dilps-producing cells (also called insulin-producing cells, IPCs) and less so DMS-producing cells. (E) Connectivity between presynaptic partners of hugin-PC neurons and mNSCs. Hugin-PC neurons share inputs with Dilps- and DMS-producing neurons but not with DH44-producing neurons. Each column across all four graphs represents a presynaptic partner of hugin-PC. Whiskers represent standard deviation.

receptor connection. The *hugin* gene encodes a prepropeptide that is post-translationally processed to produce an eight-amino-acid neuropeptide, termed pyrokinin-2 (hug-PK2) or hugin neuropeptide (*Meng et al., 2002*). This hugin neuropeptide has been shown to activate the *Drosophila* G-protein-coupled receptor (GPCR) CG8784/PK2-R1 in mammalian cell systems, but the identities of the target neurons expressing the receptor remain unknown (*Rosenkilde et al., 2003*). To address this, we used two independent methods to generate transgenic fly lines, *CG8784-GAL4::p65* and *CG8784-6kb-GAL4*, driving expression under control of putative *CG8784* regulatory sequences (*Figure 8A*; *Figure 8—figure supplement 1*). Both *CG8784-GAL4* lines drive expression of a GFP reporter in a prominent cluster of cells in the pars intercerebralis. Double stainings show that this expression co-localizes with the peptides produced by the three types of mNSCs: Dilp2, DH44, and DMS (*Figure 8B–D*; *Figure 8—figure supplement 1*). To support the receptor expression data, we performed calcium (Ca²⁺) imaging of the mNSCs upon treatment with hug-PK2 (*Figure 8—figure supplement 2*). Indeed, calcium activity of the mNSCs increased significantly after treatment with concentrations of 1 µM hug-PK2 or higher. These findings support the hypothesis that hugin-PC neurons employ both classical synaptic transmission and peptidergic signaling to target neurons of the neuroendocrine center.

Neuropeptides are produced in the soma and packaged into dense core vesicles (DCVs) before being transported to their release sites (*van den Pol, 2012*). Exploring the spatial relationship between DCVs and synapses, we observed that for both interneuron type hugin classes (hugin-PC and hugin-VNC) DCVs localized close to but not exclusively at presynaptic sites (*Figure 9A,B*). This was often the case at local swellings along the main neurites which featured multiple pre- as well as postsynaptic sites, as well as close-by DCVs. It is conceivable that such complex local synaptic circuitry might enable local peptide release. We found 98% of the synapses between hugin-PC and mNSCs to have DCVs in proximity to the presynaptic sites, opening up the possibility of co-transmission of hugin peptide and ACh (*Nusbaum et al., 2001*) (*Figure 9C*). However, most DCVs were

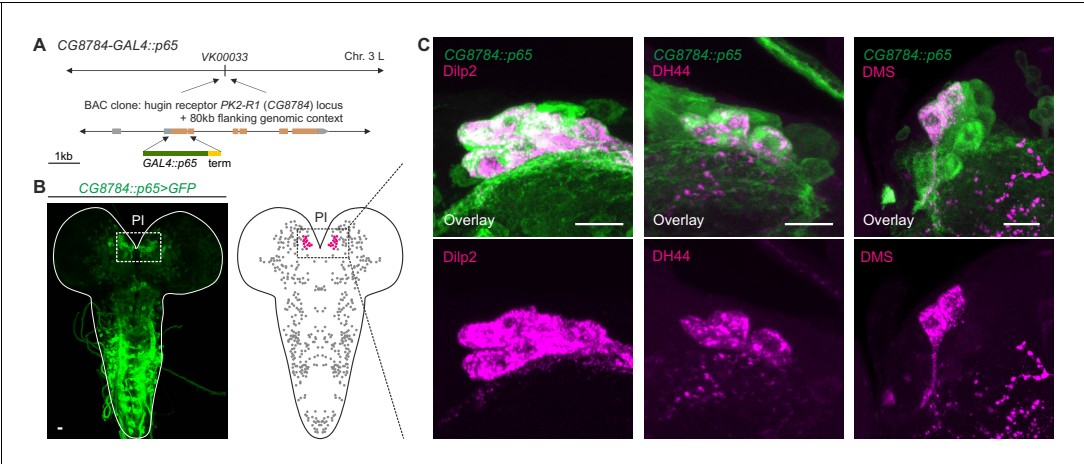

**Figure 8.** GPCR-mediated neuromodulatory transmission is used in addition to synaptic connections. (**A**) Generation and expression pattern of a hugin receptor GAL4 line, *CG8784-GAL4::p65*. Promoter-based driver line for hugin G-proteincoupled receptor PK2-R1 was generated by replacing the first coding exon of the *CG8784* loci with GAL4 in a BAC clone containing ~80 kb flanking genomic context and integrating the final BAC into *attP* site *VK00033*. (**B**) *CG8784-GAL4::p65* drives expression in cells of the pars intercerebralis (PI). (**C**) Co-staining against *Drosophila* insulin-like peptide 2 (Dilp2), diuretic hormone 44 (DH44) and Dromyosuppressin (DMS) shows that hugin receptor PK2-R1 is expressed in all median neurosecretory cells (mNSCs). Scale bars represent 10 µm.

The following figure supplements are available for figure 8:

**Figure supplement 1.** Second hugin receptor line, *CG8784-6kb-GAL4,* drives expression in median neurosecretory cells (mNSCs) of the pars intercerebralis (**PI**) similar to *CG8784-GAL4:p65*.

**Figure supplement 2.** Hugin neuropeptide increases calcium (Ca²⁺) activity in median neurosecretory cells (mNSCs).

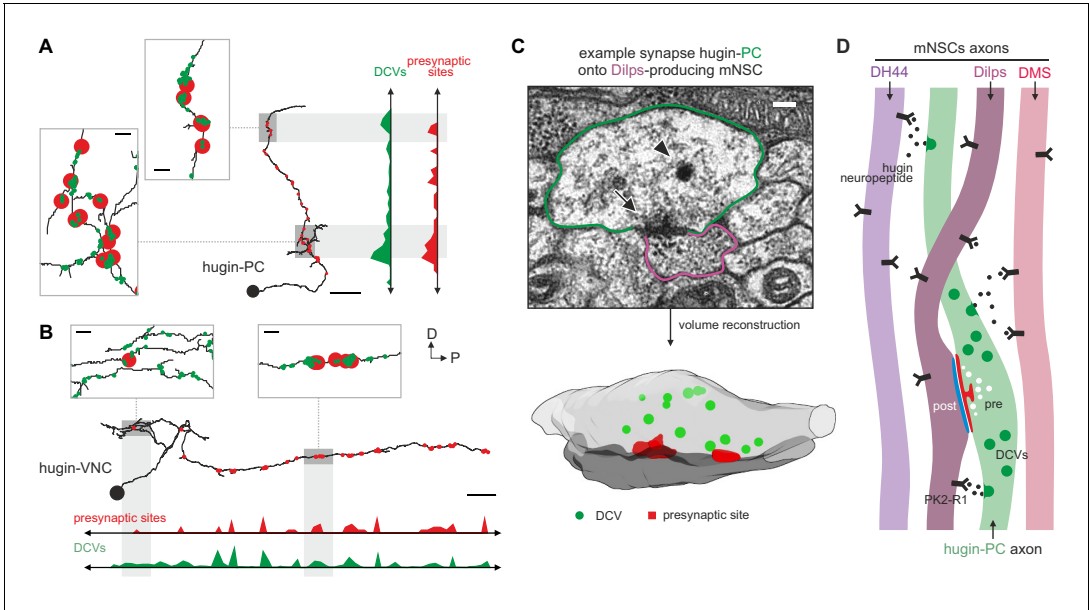

**Figure 9.** Dense core vesicles localize close to but not directly at presynaptic sites. (**A,B**), Overlay of presynaptic sites and dense core vesicles (DCVs) for exemplary hugin-PC (**A**) and hugin-VNC (**B**) neuron. DCVs are found close to but not exclusively at presynaptic sites (see inlets). Scale bars represent 10 μm (overview) and 1 μm (inlets). (**C**) Volume reconstruction of representative synapse between hugin-PC neuron and median neurosecretory cells (mNSCs) producing *Drosophila* insulin-like peptides (Dilps) shows DCVs (arrowhead) in the vicinity of presynaptic densities (arrow). Scale bar represents 100 μm. (**D**) Summarizing schematic and model. Hugin-PC neurons make classical chemical synapses almost exclusively onto Dilps-producing mNSCs. Additionally, all mNSCs express hugin receptor PK2-R1 (CG8784) and are often in close vicinity to hugin neurites, allowing para- or non-synaptically released hugin neuropeptide to bind.

probably too distant from presynaptic sites to be synaptically released, suggesting para- and non-synaptic release (*Morris and Pow, 1991*; *Maley, 1990*) (*Figure 2—figure supplement 1D*).

Taken together, these findings show that the neuroendocrine system is indeed a major downstream target of hugin neurons and that this is achieved by a combination of synaptic and GPCR-mediated neuromodulatory transmission (*Figure 9D*).

## Discussion

### Organizational principles of peptidergic microcircuits

Almost all neurons in *Drosophila* are uniquely identifiable and stereotyped (*Vogelstein et al., 2014*; *Manning et al., 2012*). This enabled us to identify and reconstruct a set of 20 peptidergic neurons in an ssTEM volume spanning an entire larval CNS (*Ohyama et al., 2015*). These neurons produce the neuropeptide hugin and have previously been grouped into four classes based on their projection targets (*Figure 2A*) (*Bader et al., 2007a*). We found that neurons of the same morphological class (a) were very similar with respect to the distribution of synaptic sites, (b) shared a large fraction of their pre- and postsynaptic partners and (c) in case of the interneuron classes (hugin-PC and hugin-VNC), neurons were reciprocally connected along their axons with other neurons of the same class. This raises the question why the CNS sustains multiple copies of morphologically very similar neurons. Comparable features have been described for a population of neurons which produce crustacean cardioactive peptide (CCAP) in *Drosophila* (*Karsai et al., 2013*). The reciprocal connections as well as the overlap in synaptic partners suggest that the activity of neurons within each interneuron class is likely coordinately regulated and could help sustain persistent activity within the population. In the mammalian pyramidal network of the medial prefrontal cortex, reciprocal connectivity between neurons is thought to contribute to the network's robustness by synchronizing activity within subpopulations and to support persistent activity (*Wang et al., 2006*). Similar interconnectivity and shared synaptic inputs have also been demonstrated for peptidergic neurons producing

gonadotropin-releasing hormone (GnRH) and oxytocin in the hypothalamus (*Campbell et al., 2009*; *Theodosis, 2002*). Likewise, this is thought to synchronize neuronal activity and allow periodic bursting.

## Functional versus connectomic map of hugin

Previous studies showed that specific phenotypes and functions can be assigned to certain classes of hugin neurons: hugin-VNC neurons increase locomotion motor rhythms but do not affect food intake, whereas hugin-PC neurons decrease food intake and are necessary for processing of aversive gustatory cues (*Schoofs et al., 2014*; *Hückesfeld et al., 2016*). For hugin-RG or hugin-PH such specific functional effects have not yet been described (for summary see *Table 1*). One conceivable scenario would be that each hugin class mediates specific aspects of an overarching 'hugin phenotype'. This would require that under physiological conditions all hugin classes are coordinately active. However, we did not find any evidence of such coordination on the level of synaptic connectivity. Instead, each hugin class forms an independent microcircuit with its own unique set of pre- and postsynaptic partners. We thus predict that each class of hugin-producing neurons has a distinct context and function in which it is relevant for the organism.

Data presented in this study provide the neural substrate for previous observation as well as open new avenues for future studies. One of the key features in hugin connectivity is the sensory input to hugin-PC, hugin-VNC and, to a lesser extent, hugin-RG. While hugin-PC neurons are known to play a role in gustatory processing, there is no detailed study of this aspect for hugin-VNC or hugin-RG neurons (*Hückesfeld et al., 2016*). Sensory inputs to hugin neurons are very heterogeneous, which suggests that they have an integrative/processing rather than a simple relay function.

Hugin neurons also have profound effects on specific motor systems: hugin-PC neurons decelerate motor patterns for pharyngeal pumping whereas hugin-VNC neurons accelerate locomotion motor patterns (*Schoofs et al., 2014*; *Hückesfeld et al., 2016*). For hugin-PC, we have demonstrated that this effect is mediated by both synaptic and hugin peptide transmissions. For hugin-VNC, this effect is independent of the hugin neuropeptide, suggesting synaptic transmission to play a key role (*Schoofs et al., 2014*). Suprisingly, we did not find any direct synaptic connections to the relevant motor neurons. However, the kinetics of the effects of hugin neurons on motor systems have not yet been studied at a high enough temporal resolution (i.e. by intracellular recordings) to assume monosynaptic connections. It is thus well conceivable that connections to the respective motor systems are polysynaptic and occur further downstream. Alternatively, this may involve an additional non-synaptic (peptidergic) step. A strong candidate for this is the neuroendocrine system which we identify as the major downstream target of hugin-PC neurons. Among the endocrine targets of hugin, the insulin-producing cells (IPCs) have long been known to centrally regulate feeding behavior (*Erion and Sehgal, 2013*). It is not known if insulin-signaling directly affects motor patterns in *Drosophila*. Nevertheless, increased insulin signaling has strong inhibitory effects on food-related sensory processing and feeding behavior (*Wu et al., 2005a*; *Wu et al., 2005b*). Whether the neuroendocrine system is a mediator of the suppressive effects of hugin-PC neurons on food intake remains to be determined.

**Table 1.** Summary of known effects of hugin classes and their connectivity.

| hugin class | known effects | connectivity |
|---|---|---|
| PC | decrease food intake*; decelerate AN motor pattern (for pharyngeal pumping)*; necessary for bitter avoidance* | chemosensory input via AN; output onto neuroendocrine system; unidentified interneuron inputs and outputs in SEZ and higher brain centers |
| VNC | accelerate M6 motor patterns (for locomotion)[†] | chemosensory input via AN; unknown sensory input from abdominal nerves; unidentified interneuron inputs in SEZ; outputs in VNC |
| RG | unknown | weak chemosensory input via AN; inputs from unidentified interneurons in SEZ; no synaptic outputs in CNS |
| PH | unknown | inputs from unidentified interneurons in SEZ; no synaptic outputs in CNS |

Known effects based on *****Hückesfeld et al. (2016)** and [†]***Schoofs et al. (2014)**. AN, antennal nerve; SEZ, subesophageal zone; VNC, ventral nerve cord.

The first functional description of hugin in *Drosophila* was done in larval and adult (*Melcher and Pankratz, 2005*), while more recent publications have focused entirely on the larva (*Schoofs, 2014*; *Hückesfeld et al., 2016*). One of the main reasons for this is the smaller behavioral repertoire of the larva: the lack of all but the most fundamental behaviors makes it well suited to address basic questions. Nevertheless, it stands to reason that elementary circuits should be conserved between larval and adult flies. To date, there is no systematic comparison of hugin across the life cycle of *Drosophila*. However, there is indication that hugin neurons retain their functionality from larva to the adult fly. First, morphology of hugin neurons remains virtually the same between larva and adults (*Melcher and Pankratz, 2005*). Second, hugin neurons seem to serve similar purposes in both stages: they acts as a brake on feeding behavior – likely as response to aversive sensory cues (*Hückesfeld et al., 2016*). In larvae, artificial activation of this brake shuts down feeding (*Schoofs et al., 2014*). In adults, removal of this break by silencing of hugin neurons leads to a facilitation (earlier onset) of feeding (*Melcher and Pankratz, 2005*). Such conservation of neuropeptidergic function between larval and adult *Drosophila* has been observed only in a few cases. Prominent examples are short (*Lee et al., 2004*, *2008*) and long neuropeptide F (*Beshel and Zhong, 2013*; *Wang et al., 2013*), both of which show strong similarities with mammalian NPY. The lack of additional examples is not necessarily due to actual divergence of peptide function but rather due to the lack of data across both larva and adult. Given the wealth of existing data on hugin in larvae, it would be of great interest to investigate whether and to what extent the known features (connectivity, function, etc.) of this system are maintained throughout *Drosophila*'s life history.

## Parallel synaptic and neuromodulatory connections along chemosensory-endocrine axis

A neural network is a highly dynamic structure and is subject to constant change, yet it is constrained by its connectivity and operates within the framework defined by the connections made between its neurons (*Getting, 1989*). On one hand, this connectivity is based on anatomical connections formed between members of the network, namely synapses and gap junctions. On the other hand, there are non-anatomical connections that do not require physical contact due to the signaling molecules, such as neuropeptides/-hormones, being able to travel considerable distances before binding their receptors (*van den Pol, 2012*). Our current integrated analysis of the operational framework for a set of neurons genetically defined by the expression of a common neuropeptide, positions hugin-producing neurons as a novel component in the regulation of neuroendocrine activity and the integration of sensory inputs. We show that most hugin neurons receive chemosensory input in the subesophageal zone, the brainstem analog of *Drosophila* (*Ghysen, 2003*; *Schoofs et al., 2014*). Of these, one class is embedded into a network whose downstream targets are median neurosecretory cells (mNSCs) of the pars intercerebralis, a region homologous to the mammalian hypothalamus (*Hartenstein, 2006*). We found that hugin neurons target mNSCs by two mechanisms. First, by classic synaptic transmission as our data strongly suggest that acetylcholine (ACh) acts as transmitter at these synapses. Accordingly, subsets of mNSCs have been shown to express a muscarinic ACh receptor (*Cao et al., 2014*). Whether additional ACh receptors are expressed is unknown. Second, by non-anatomical, neuromodulatory transmission using a peptide-receptor connection, as demonstrated by the expression of hugin G-protein-coupled receptor PK2-R1 (*CG8784*) in mNSCs. Strikingly, while PK2-R1 is expressed in all mNSCs, the hugin neurons have many synaptic contacts onto insulin-producing cells but few to DMS and DH44 neurons. This mismatch in synaptic vs. peptide targets among the mNSCs suggests an intricate influence of hugin-producing neurons on this neuroendocrine center. In favor of a complex regulation is that those mNSCs that are synaptically connected to hugin neurons additionally express a pyrokinin-1 receptor (PK1-R, *CG9918*) which, like PK2-R1, is related to mammalian neuromedinU receptors (*Alfa et al., 2015*; *Cazzamali et al., 2005*; *Park et al., 2002*). There is some evidence that PK1-R might also be activated by the hugin neuropeptide, which would add another regulatory layer (*Cazzamali, 2005*).

The concept of multiple messenger molecules within a single neuron is well established and appears to be widespread among many organisms and neuron types (*Burnstock, 2004*; *Nusbaum et al., 2001*; *Merighi, 2002*; *Brezina, 2010*; *Li and Kim, 2008*). For example, cholinergic transmission plays an important role in mediating the effect of NMU in mammals. This has been demonstrated in the context of anxiety but not yet for feeding behavior (*Telegdy and Adamik, 2013*; *Tanaka and Telegdy, 2014*). There are, however, only few examples of simultaneous

employment of neuromodulation and fast synaptic transmission in which specific targets of both messengers have been investigated at single-cell level. In many cases, targets and effects of classic and peptide co-transmitters seem to diverge (e.g. (*Sun et al., 2003*; *Li and van den Pol, 2006*; *Stein et al., 2007*). In contrast, AgRP neurons in the mammalian hypothalamus employ neuropeptide Y, the eponymous agouty-related protein (AgRP) and the small molecule transmitter GABA to target pro-opiomelanocortin (POMC) neurons in order to control energy homeostasis (*Cansell et al., 2012*). Also, reminiscent of our observations is the situation in the frog sympathetic ganglia, where preganglionic neurons use both ACh and a neuropeptide to target so-called C cells but only the neuropeptide additionally targets B cells. In both targets, the neuropeptide elicits late, slow excitatory postsynaptic potentials (EPSPs) (*Jan and Jan, 1983*). It is conceivable that hugin-producing neurons act in a similar manner by exerting a slow, lasting neuromodulatory effect on all mNSCs and a fast, transient effect exclusively on synaptically connected mNSCs. Alternatively, the hugin neuropeptide could facilitate the postsynaptic effect of acetylcholine. Such is the case in *Aplysia* where a command-like neuron for feeding employs acetylcholine and two neuropeptides, feeding circuit activating peptide (FCAP) and cerebral peptide 2 (CP2). Both peptides work cooperatively on a postsynaptically connected motor neuron to enhance EPSPs in response to cholinergic transmission (*Koh et al., 2003*).

In addition to the different timescales that neuropeptides and small molecule transmitters operate on, they can also be employed under different circumstances. It is commonly thought that low-frequency neuronal activity is sufficient to trigger fast transmission using small molecule transmitters, whereas slow transmission employing neuropeptides requires higher frequency activity (*Nusbaum et al., 2001*). Hugin-producing neurons could employ peptidergic transmission only as a result of strong excitatory (e.g. sensory) input. There are, however, cases in which base activity of neurons is already sufficient for graded neuropeptide release: *Aplysia* ARC motor neurons employ ACh as well as neuropeptides and ACh is generally released at lower firing rates than the neuropeptide. This allows the motor neuron to function as purely cholinergic when firing slowly and as cholinergic/peptidergic when firing rapidly (*Whim and Lloyd, 1989*). However, peptide release already occurs at the lower end of the physiological activity of those neurons (*Weiss et al., 1993*; *Vilim et al., 1996*). It remains to be seen how synaptic and peptidergic transmission in hugin neurons relate to each other.

The present study is one of very few detailed descriptions of differential targets of co-transmission and – to our knowledge – the first of its kind in *Drosophila*. We hope these findings in a genetically tractable organism will provide a basis for elucidating some of the intriguing modes of action of peptidergic neurons.

## Comparative view of hugin and neuromedin systems

The mammalian homolog of hugin, neuromedinU (NMU), is found in the CNS as well as in the gastrointestinal tract (*Ballesta et al., 1988*). Its two receptors, NMUR1 and NMUR2, show differential expression. NMUR2 is abundant in the brain and the spinal cord, whereas NMUR1 is expressed in peripheral tissues, in particular in the gastrointestinal tract (*Mitchell et al., 2009*). Both receptors mediate different effects of NMU. The peripheral NMUR1 is expressed in pancreatic islet β cells in humans and allows NMU to potently suppress glucose-induced insulin secretion (*Alfa et al., 2015*). The same study also showed that Limostatin (Lst) is a functional homolog of this peripheral NMU in *Drosophila*: Lst is expressed by glucose-sensing, gut-associated endocrine cells and suppresses the secretion of insulin-like peptides. The second, centrally expressed NMU receptor, NMUR2, is necessary for the effect of NMU on food intake and physical activity (*Zeng et al., 2006*; *Peier et al., 2009*). In this context, NMU is well established as a factor in regulation of the hypothalamo-pituitary axis (*Wren et al., 2002*; *Malendowicz et al., 2012*) and has a range of effects in the hypothalamus, the most important being the release of corticotropin-releasing hormone (CRH) (*Hanada et al., 2001*, *2003*). We show that a subset of hugin-producing neurons targets the pars intercerebralis, the *Drosophila* homolog of the hypothalamus, in a similar fashion: neuroendocrine target cells in the pars intercerebralis produce a range of peptides, including diuretic hormone 44 which belongs to the insect CRH-like peptide family (*Cabrero et al., 2002*) (*Figure 10*). Given these similarities, we propose that hugin is homologous to central NMU just as Lst is a homologous to peripheral NMU. Demonstration that central NMU and hugin circuits share similar features beyond targeting

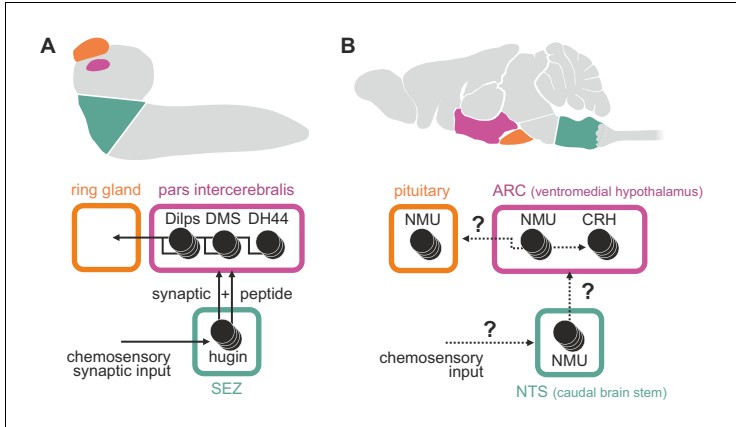

**Figure 10.** Summary of hugin connectivity and hypothetical implications for neuromedinU in mammals. (**A**) Hugin neurons link chemosensory neurons that enter the subesophageal zone (SEZ) and neuroendocrine cells of the pars intercerebralis by synaptic as well as peptide-receptor connections. (**B**) Distribution of NMU-positive neurons in mammals is much more complex. The effect of neuromedinU (NMU) on feeding and physical activity originates in the arcuate nucleus (ARC) of the hypothalamus where it causes release of corticotropin-releasing hormone (CRH) which itself is a homolog of diuretic hormone 44 (DH44) in *Drosophila*. NMU-positive neurons have also been found in the nucleus of the solitary tract (NTS) a chemosensory center in the caudal brain stem. It remains to be seen if, similar to hugin neurons, NMU neurons serve as a link between chemosensory and neuroendocrine system.

neuroendocrine centers, e.g. the integration of chemosensory inputs, will require further studies on NMU regulation and connectivity.

Previous work on vertebrate and invertebrate neuroendocrine centers suggests that they evolved from a simple brain consisting of cells with dual sensory/neurosecretory properties, which later diversified into optimized single-function cells (*Tessmar-Raible et al., 2007*). There is evidence that despite the increase in neuronal specialization and complexity, connections between sensory and endocrine centers have been conserved throughout evolution (*Yoon et al., 2005*; *Strausfeld 2012*; *Abitua et al., 2015*). We propose that the connection between endocrine and chemosensory centers provided by hugin neurons represents such a conserved circuit that controls basic functions like feeding, locomotion, energy homeostasis and sex.

Indisputably, the NMU system in mammals is much more complex as NMU is found more widespread within the CNS and almost certainly involves a larger number of different neuron types. This complexity, however, only underlines the use of numerically smaller nervous systems such as *Drosophila*'s to generate a foundation to build upon. Moreover, NMU/NMU-like systems may have similar functions not just in mammals and *Drosophila* but also other vertebrates such as fish (*Chiu et al., 2016*; *Li et al., 2015*) and other invertebrates such as *C. elegans* (*Maier et al., 2010*). In summary, our findings should encourage research in other organisms, such as the involvement of NMU and NMU homologs in relaying chemosensory information onto endocrine systems, and more ambitiously, to elucidate their connectomes in order to allow comparative analyses of the underlying network architecture.

## Materials and methods

### Neuronal reconstruction

Reconstructions were based on an ssTEM (serial section transmission electron microscope) data set comprising an entire central nervous system and the ring gland of a first-instar *Drosophila* larva. Generation of this data set was described previously (*Ohyama et al., 2015*). Neurons' skeletons were manually reconstructed using a modified version of CATMAID (http://www.catmaid.org) (*Saalfeld et al., 2009*). Hugin-PH (pharynx) neurons were first identified by reconstructing all axons in the prothoracic accessory nerve, through which these neurons exit the CNS toward the pharynx.

Similarly, hugin-RG (ring gland) neurons were identified by reconstructing all neurosecretory cells that target the ring gland. To find the remaining hugin neurons, neighbors of already identified hugin neurons were reconstructed. Among those, the remaining hugin neurons were unambiguously identified based on previously described morphological properties such as projection targets, dendritic arborizations, relative position to each other and prominent landmarks like antennal lobes or nerves (*Bader et al., 2007a*, *Bader et al., 2007b*). The mapped synaptic connections represent fast, chemical synapses matching previously described typical criteria: thick black active zones, pre- (e.g. T-bar, vesicles) and postsynaptic membrane specializations (*Prokop and Meinertzhagen, 2006*). Hugin inputs and outputs were traced by following the pre- and postsynaptically connected neurites to the respective neurons' somata or nerve entry sites in sensory axons. Subsequently, all sensory and endocrine neurons synaptically connected to hugin neurons were fully reconstructed. Interneurons were fully reconstructed if (a) homologous neurons were found in both hemispheres/-segments (did not apply to medially unpaired neurons) and (b) at least one of the paired neurons was connected by a minimum of three synapses to/from hugin neurons. Neurons that did not fit either criterion were not fully reconstructed and thus excluded from statistical analysis. This resulted in the reconstruction 177 synaptic partners that together covered 90%/96% of hugin neurons' above threshold pre-/postsynaptic sites (*Figure 5—figure supplement 1*). The same parameters were applied to the reconstruction of synaptic partners of median neurosecretory cells (mNSCs). Morphological plots and example synapse's volume reconstruction were generated using custom python scripts or scripts for Blender 3D (www.blender.org). The script for a CATMAID-Blender interface is on Github (https://github.com/schlegelp/CATMAID-to-Blender). See supplemental neuron atlas (*Supplementary files 1,2*) of all reconstructed neurons and their connectivity with hugin neurons.

### Normalized connectivity similarity score

To compare connectivity between neurons (*Figure 5B*), we used a modified version of the similarity score described by Jarrell et al. (*Jarrell et al., 2012*):

$$f\left(A_{ik}, A_{jk}\right) = min\left(A_{ik}, A_{jk}\right) - C_1 max\left(A_{ik}, A_{jk}\right) e^{-C_2 min\left(A_{ik}, A_{jk}\right)}$$

With the overall connectivity similarity score for vertices i and j in adjacency matrix A being the sum of $f\left(A_{ik}, A_{jk}\right)$ over all connected partners k. $C_1$ and $C_2$ are variables that determine how similar two vertices have to be and how negatively a dissimilarity is punished. Values used were: $C_1 = 0.5$ and $C_2 = 1$. To simplify graphical representation, we normalized the overall similarity score to the minimal (sum of $-C_1 max\left(A_{ik}, A_{jk}\right)$ over all k) and maximal (sum of $max\left(A_{ik}, A_{jk}\right)$ over all k) achievable values, so that the similarity score remained between 0 and 1. Self-connections ($A_{ii}, A_{jj}$) and $A_{ij}$ connections were ignored.

### Synapse similarity score

To calculate similarity of synapse placement between two neurons, we calculated the synapse similarity score (*Figure 6D*):

$$f(i_s, j_k) = e^{\frac{-d_{sk}^2}{2\sigma^2}} e^{-\frac{|n(i_s) - n(j_k)|}{n(i_s) + n(j_k)}}$$

With the overall synapse similarity score for neurons i and j being the average of $f(i_s, j_k)$ over all synapses s of i. Synapse k being the closest synapse of neuron j to synapses s [same sign (pre-/postsynapse) only]. $d_{sk}$ being the linear distance between synapses s and k. Variable $\sigma$ determines which distance between s and k is considered as close. $n(j_k)$ and $n(i_s)$ are defined as the number of synapses of neuron j/i that are within a radius $\omega$ of synapse k and s, respectively (same sign only). This ensures that in case of a strong disparity between $n(i_s)$ and $n(j_k)$, $f(i_s, j_k)$ will be close to zero even if distance $d_{sk}$ is very small. Values used: $\sigma = \omega = 2000$ nm.

### Clustering

Clusters for dendrograms were created based on the mean distance between elements of each cluster using the average linkage clustering method. Clusters were formed at scores of 0.2 for synapse similarity score (*Figure 6B,E*) and 0.4 for connectivity similarity score (*Figure 7D*).

## Percentage of synaptic connections

Percentage of synaptic connections was calculated by counting the number of synapses that constitute connections between neuron A and a given set of pre- or postsynaptic partners (e.g. sensory neurons) divided by the total number of either incoming or outgoing synaptic connections of neuron A. For presynaptic sites, each postsynaptic neurite counted as a single synaptic connection.

## Statistics

Statistical analysis was performed using custom Python scripts; graphs were generated using Sigma Plot 12.0 (www.sigmaplot.com) and edited in Adobe Corel Draw X5 (www.corel.com).

## Generation of hugin receptor *CG8784* promoter lines

The *CG8784-GAL4::p65* construct (*Figure 8*) was created using recombineering techniques (*Warming et al., 2005*) in P[acman] bacterial artificial chromosome (BAC) clone CH321-45L05 (*Venken et al., 2009*) (obtained from Children's Hospital Oakland Research Institute, Oakland, CA), containing *CG8784* within ~80 kb of flanking genomic context. A generic landing-site vector was created by flanking the kanamycin-resistance/ streptomycin-sensitivity marker in *pSK+-rpsL-kana* (*Wang et al., 2009*) (obtained from AddGene.org, plasmid #20871) with 5 'and 3' homology arms (containing *GAL4* coding sequences and *HSP70* terminator sequences, respectively) amplified from *pBPGUw* (*Pfeiffer et al., 2008*). *CG8784*-specific homology arms were added to this cassette by PCR using the following primers (obtained as Ultramers from Integrated DNA Technologies, Inc., Coralville, Iowa; the lower case portions are *CG8784*-specific targeting sequences, and the capitalized portions match the *pBPGUw* homology arms):

| | |
|---|---|
| CG8784::p65-F | tggcgtggcgtggagtggatagagtccacaattaatcga cgacagctagtATGAAGCTACTGTCTTCTATCGAACAAGC |
| CG8784::p65-R | tttgccgcattacgcatacgcaatggtgtccctcaaaaa tgccatctcacGATCTAAACGAGTTTTTAAGCAAACTCACTCCC |

This cassette was recombined into the BAC, replacing the coding portion of the first coding exon, and then full-length *GAL4::p65-HSP70* amplified from *pBPGAL4.2::p65Uw* (*Pfeiffer et al., 2010*) was recombined into the landing site in a second recombination. Introns and exons following the insertion site were retained in case they contain expression-regulatory sequences, although they are presumably no longer transcribed. Correct recombination was verified by sequencing the recombined regions, and the final BAC was integrated into the third-chromosome *attP* site *VK00033* (*Venken et al., 2006*) by Rainbow Transgenic Flies, Inc. (Camarillo, CA).

The *CG8784-6kb-GAL4* (*Figure 8—figure supplement 1*) was created using standard restriction-digestion/ligation techniques in pCaSpeR-AUG-Gal4-X vector (*Vosshall et al., 2000*). An approximately 6 kb promoter fragment 5' of the first coding exon was amplified using the following primers and inserted into a pCaSpeR vector (Addgene.org, plasmid #8378) containing a start codon (AUG) and the *GAL4* gene (*Figure 8—figure supplement 1*).

| | |
|---|---|
| CG8784-6kb-F | AATATCTTGGCAACGAAGTCC |
| CG8784-6kb-R | AGCTGTCGTCGATTAATTGTG |

This construct was integrated into the genome via P-element insertion.

## Immunohistochemistry

For antibody stainings of *CG8784-GAL4::p65*, larvae expressing *JFRC2-10xUAS-IVS-mCD8::GFP* (*Pfeiffer et al., 2010*) driven by *CG8784-GAL4::p65* were dissected in PBS. Brains were fixed in 4% formaldehyde in PBS for 1 hr, rinsed, blocked in 5% normal goat serum, and incubated overnight at 4°C with primaries: sheep anti-GFP (AbD Serotec #4745–1051), 1:500; rabbit anti-DH44 (*Cabrero et al., 2002*) (gift of Jan Veenstra), 1:1000; rabbit anti-DILP2 (*Veenstra et al., 2008*) (gift

of Jan Veenstra), 1:1000; 1:1000; and rabbit anti-DMS (*Schoofs et al., 1993*; *Park et al., 2008*) (gift of Luc van den Bosch and Liliane Schoofs), 1:500. Tissues were rinsed and incubated overnight at 4°C in secondaries: Alexa Fluor 488 donkey anti-sheep (Jackson ImmunoResearch, #713–545-147) and rhodamine red-X donkey anti-rabbit (Jackson ImmunoResearch #711–296-152), both 1:500. Brains were rinsed and dehydrated through an ethanol-xylene series, mounted in DPX, and scanned on a Zeiss LSM 510 confocal microscope.

For antibody stainings of *CG8784-6kb-GAL4*, larvae expressing *10XUAS-mCD8::GFP* (Bloomington, #32184) driven by *CG8784-6kb-GAL4* were dissected in PBS. Brains were fixed in 4% paraformaldehyde for 30 min, rinsed, blocked in 5% normal goat serum, and incubated overnight at 4°C with primaries: goat anti-GFP-FITC (abcam, ab26662), 1:500; rabbit anti-DH44 (*Cabrero et al., 2002*) (gift of Jan Veenstra), 1:1000; guinea pig anti-Dilp2 (*Bader et al., 2013*) (Pankratz lab), 1:500 and rabbit anti-DMS (*Schoofs et al., 1993*; *Park et al., 2008*) (gift of Luc van den Bosch and Liliane Schoofs), 1:500. Tissues were rinsed and incubated overnight at 4°C in secondaries: anti-rabbit Alexa Fluor 633 (Invitrogen, A-21070) and anti-guinea pig Alexa Fluor 568 (Invitrogen, A-11075), both 1:500. Brains were rinsed, mounted in Mowiol (Roth, 0713), and scanned on a Zeiss LSM 710 confocal microscope.

For antibody stainings against choline acetyltransferase (ChAT), larvae expressing a YFP-tagged halorhodopsin (UAS- eNpHR-YFP; Bloomington, #41753) driven by HugS3-GAL4 (*Melcher and Pankratz, 2005*) as marker were prepared following the above protocol for *CG8784-6kb-GAL4* stainings. Primary antibodies used: goat anti-GFP-FITC (abcam, ab26662), 1:500; mouse anti-ChAT (Developmental Studies Hybridoma Bank, ChAT4B1) (*Takagawa and Salvaterra, 1996*), 1:1000. Secondary antibodies used: anti-mouse Alexa Fluor 633 (Invitrogen, A-21046).

For investigation of ChAT promoter activity in hugin neurons, larvae expressing UAS-cd8a::mRFP (Bloomington, #27399) under the control of ChAT-GAL4 7.4 kb (Bloomington, #6798) and YFP directly under the control of the hugin promoter (hug-YFP; (*Melcher and Pankratz, 2005*) were prepared following the above protocol for *CG8784-6kb-GAL4* stainings. Primary antibodies used: goat anti-GFP-FITC (abcam, ab26662), 1:500; mouse anti-RFP (abcam, ab65856), 1:500. Secondary antibodies used: anti-mouse Alexa Fluor 633 (Invitrogen, A-21046).

For quantification of ChAT antibody signals/ChAT promoter activity, samples were scanned on a Zeiss LSM 710 confocal microscope using a 63X objective (Zeiss). Settings were kept the same over all scans. Regions of interest were placed through the center of each hugin neuron's soma, and the mean intensity was measured using ImageJ (https://imagej.nih.gov/ij/index.html) (*Schneider et al., 2012*). Hugin-PC and hugin-RG neurons were identified based on soma position and morphology. Hugin-VNC and hugin-PH could not be unambiguously discriminated as they were usually too tightly clustered. They were thus treated as a single group. For background normalization, an approximately 10×10 μm rectangle from the center of the image stack was chosen.

## RNAi experiments

To investigate the role of acetylcholine as transmitter of hugin neurons, food intake and electrophysiological experiments were performed. Experimental procedures, materials and setups used in these assays been described extensively in *Schoofs et al. (2014)*. The hugin and ChAT RNAi experiments presented in *Figure 3* were performed together as part of a larger screen on neuronal populations and genes involved in larval feeding behavior. A portion of this screen, which did not include the ChAT RNAi data that we are now presenting here, was published previously in *Schoofs et al. (2014)*. In the following, we briefly summarize procedures of *Schoofs et al. (2014)* for reader convenience. Please see that reference for more detailed description.

The following GAL4 driver and UAS effector lines were used: HugS3-GAL4 (*Melcher and Pankratz 2005*), UAS-dTrpA1 (Bloomington, #26263), UAS-LacZRNAi (gift from M. Jünger), UAS-HugR-NAi1A (*Schoofs et al., 2014*) and UAS-ChAT-RNAi (TriP.JF01877) (Bloomington, #25856) (*Barnstedt et al., 2016*; *Plaçais et al., 2013*). OregonR and OregonR x UAS-dTrpA1 were used as control.

For the food intake assay, third instar larvae were first washed and starved for 30 min on RT. They were then transferred on yeast paste colored with crimson red and allowed to feed for 20 min. Experiments were performed at 32°C for dTrpA1-induced activation of hugin neurons and at 18°C as control condition. Afterwards larvae were photographed and the amount of food ingested was calculated as the area of the alimentary tract stained by the colored yeast divided by body surface area

using ImageJ (https://imagej.nih.gov/ij/index.html) (*Schneider et al., 2012*). Data are represented as fold change between control condition (18°C) and dTrpA1-induced activation (32°C) normalized to the control.

For the electrophysiological assay, semi-intact preparations of third instar larvae were made in saline solution (*Rohrbough and Broadie, 2002*). E*n passant* extracellular recordings of the antennal nerve (AN) were performed following previously described protocol (*Schoofs et al., 2014*). During the recordings, temperature of the CNS was alternated between 18°C (control condition) and 32°C (dTrpA1 activation). For analysis, fictive motor patterns of the pharyngeal pump (also: cibarial dilator musculature, CDM) were analyzed: fold change in cycle frequency between pairs of successive 18°C and 32°C sections of a recording was calculated.

### Pharmacological experiments and calcium ($Ca^{2+}$) imaging

Hugin-derived pyrokinin 2 (hug-PK2) was synthesized by Iris Biotech (Marktredwitz, Germany) using the amino acid sequence SVPFKPRL-NH2. The C terminus was amidated. The effect of hug-PK2 on calcium activity in median neurosecretory cells (mNSCs) was investigated using the calcium integrator CaMPARI (*Fosque et al., 2015*). To drive expression of CaMPARI in mNSCs, *CG8784-6kb-GAL4* flies were crossed to UAS-CaMPARI (Bloomington #58761). Larval brains were dissected and placed in saline solution (*Rohrbough and Broadie, 2002*) containing either no, 100 nM, 1 µM or 10 µM hug-PK2. After 1 min of incubation, 405 nm photoconversion light was applied for 15 s. Afterwards, brains were placed on a poly-l-lysine-coated (Sigma-Aldrich, P8920) cover slide and scanned using a Zeiss LSM 780 confocal microscope. Settings were kept the same over all scans. Calcium activity was calculated as the ratio of the fluorescence of photoconverted (red) to unconverted (green) CaMPARI using ImageJ.

## Acknowledgements

We thank Jan Veenstra and Liliane Schoofs for their gifts of antisera, and Hubert Amrein, Ron Tanimoto, Leslie Vosshall, Christian Jünger, Barret Pfeiffer and Gerry Rubin for plasmids and fly lines. We thank SFB 645 and 704, DFG Cluster of Excellence ImmunoSensation and DFG grant PA 787 for financial support. We thank the Fly EM Project Team at HHMI Janelia for the gift of the EM volume, the HHMI visa office, and HHMI Janelia for funding. We also thank Lucia Torres, Gaia Tavosanis, Gáspár Jékely, Gregory Jefferis, Ingo Zinke, Scott Sternson, Christian Wegener, Volker Hartenstein and Nicholas Strausfeld for critical comments on earlier versions of this manuscript. The EM image data is available via the Open Connectome Project (http://www.openconnectomeproject.org).

## Additional information

### Funding

| Funder | Author |
|---|---|
| Howard Hughes Medical Institute | Michael J Texada<br>Casey M Schneider-Mizell<br>Haluk Lacin<br>Feng Li<br>Richard D Fetter<br>James W Truman<br>Albert Cardona |
| Deutsche Forschungsgemeinschaft | Philipp Schlegel<br>Anton Miroschnikow<br>Andreas Schoofs<br>Sebastian Hückesfeld<br>Marc Peters<br>Michael J Pankratz |

The funders had no role in study design, data collection and interpretation, or the decision to submit the work for publication.

## Author contributions
PS, AS, AC, Conception and design, Acquisition of data, Analysis and interpretation of data, Drafting or revising the article; MJT, Acquisition of data, Analysis and interpretation of data, Drafting or revising the article; AM, HL, FL, Acquisition of data; SH, Conception and design, Acquisition of data, Analysis and interpretation of data; MP, CMS-M, RDF, Acquisition of data, Analysis and interpretation of data; JWT, Analysis and interpretation of data, Drafting or revising the article; MJP, Conception and design, Analysis and interpretation of data, Drafting or revising the article

## Author ORCIDs
Philipp Schlegel, http://orcid.org/0000-0002-5633-1314
Michael J Texada, http://orcid.org/0000-0003-2479-1241
Casey M Schneider-Mizell, http://orcid.org/0000-0001-9477-3853
Haluk Lacin, http://orcid.org/0000-0003-2468-9618
Albert Cardona, http://orcid.org/0000-0003-4941-6536
Michael J Pankratz, http://orcid.org/0000-0001-5458-6471

## Additional files

### Supplementary files
• Supplementary file 1. PDF Neuron Atlas - Morphology and connectivity of reconstructed neurons. Reconstructions of (A) hugin-PC, (B) hugin-VNC, (C) hugin-RG, (D) hugin-PH neurons, (E) insulin-producing cells (IPCs), (F) DH44-producing cells, (G) DMS-producing cells, (H) antennal nerve (AN) sensory neurons as clustered in *Figure 6,* (I) abdominal nerve sensory neurons, (J) paired interneurons and (K) unpaired medial interneurons. A dorsal view of each cell is shown on the left, and a frontal view on the right. Neuron ids (e.g. #123456) are provided to allow comparison between PDF and Blender atlas. Outline of the nervous system and the ring gland are shown in grey and dark grey, respectively. Table shows number of synapses of given neurons onto (left) and from (right) the hugin neuron represented in that row. Neurons are displayed as corresponding pairs of the left/right hemisegment with the exception of sensory neurons and unpaired medial interneurons.

• Supplementary file 2. Blender 3D Neuron Atlas – Morphology of reconstructed neurons as Blender file. To view, please download Blender (www.blender.org). Reconstructed neurons are sorted into layers: hugin neurons (1), mNSCs (2), sensory neurons (3), interneurons (4) and mesh of the larval brain (5, hidden by default). Neuron names contain id (e.g. #123456) to allow comparison between Blender and PDF atlas. Neurons have been resampled by a factor of four to reduce vertex count. 1 nm = 0.0001 Blender units.

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
