## [Decision Letter]

Thank you for submitting your article "Synaptic transmission parallels neuromodulation in a central food-intake circuit" for consideration by *eLife*. Your article has been reviewed by three peer reviewers, and the evaluation has been overseen by a Reviewing Editor, Ronald L. Calabrese, and Eve Marder as the Senior Editor. The reviewers have opted to remain anonymous.

The reviewers have discussed the reviews with one another and the Reviewing Editor has drafted this decision to help you prepare a revised submission.

Summary:

The paper by Schlegel et al. describes the circuitry of hugin-expressing neurons in the *Drosophila* larva. The neuron reconstructions and the analysis of connectivity are an impressive body of work. The authors combined it with the detailed analysis of the chemical connectivity of one class of hugin-producing neurons and their targets expressing the GPCR for hugin. The possibility of directly correlating individual neurons in light microscopy samples to the EM volume allowed the analysis of such molecular connectivity in the circuit at a cellular resolution. The 20 hugin neurons make up 4 distinct microcircuits: the activity of each of these is likely to underlie distinct features of feeding behavior.

Essential revisions:

There are some reservations about this paper that we ask the authors to address in revision.

1) The 4 hugin microcircuits likely have different functions, but are there any indications of what these might be? Can the authors provide any evidence that might begin to identify these functions? Is there evidence supporting a feeding-related function for each of the 4 hugin modules? Commonly, neuropeptides (like other transmitter molecules) subserve neural signaling in numerous unrelated contexts.

2) There is concern that the homology/analogy with the mammalian NMU neuron is overemphasized. Quoting from one of the expert reviews, "The authors compare the circuitry of hugin neurons to the connectivity of NMU neurons in vertebrates. Homologizing neuron types and brain regions across large evolutionary distances is difficult. The authors are apparently aware of these difficulties and sometimes use the term homology and sometimes analogy. These are of course not the same.

One criterion for establishing homologous neuron types across phyla would be that the neurons express a similar set of transcription factors and effector genes and are located in brain areas that as regions are possibly homologous (e.g., the spinal cord and the VNC).

The difficulty in comparing the fly hugin neurons to the NMU neurons in vertebrates is that in vertebrates the expression pattern of NMU is more complex. NMU neurons can be found in the hypothalamus, the pituitary, the brainstem, the spinal cord, and throughout the gastrointestinal tract. However, in the fly, hugin+ neurons are only found in the SEZ. These project to the VNC, PC and RG, but there are no hugin+ somata in these regions. The authors compare the projections of the hugin neurons to the distinct sets of NMU+ neurons in vertebrates. This is problematic. It is only the hugin+ cells in the SEZ in *Drosophila* that may be comparable to NMU+ cells in the brainstem in vertebrates. In this sense, Figure 1 is misleading. The authors must be more careful about this and explain more carefully what they are trying to compare to what and based on which criteria. Regarding a comparison of circuitry, it is even more problematic, since not much is known about the circuitry of brainstem NMU neurons in vertebrates.

The similarity in chemical connectivity of hugin/NMU-DH44/CRF neurons between the fly and vertebrates is intriguing. However, it is again difficult to directly compare these circuits, since there are no hugin+ neurons in the pars intercerebralis that would be equivalent to the hypothalamic NMU neurons."

3) Co-transmission as a theme is only partially supported and weakly contextualized. Quoting from one of the expert reviewers, "The conclusions regarding the functional consequences of the co-localized (not yet identified) small molecule transmitter and hugin peptide are only likely or possible ones, not firm ones. For example, the extent to which the influence of the hugin neurons, and particularly that of any one of the 4 modules, mimics the previously documented behavioral consequences of genetically manipulating the presence of the hugin peptide remains to be determined. Also, the proposed target neurons of neuronally released hugin peptide are the likely but not necessarily actual target neurons, despite the presynaptic presence of hugin peptide and postsynaptic presence of hugin receptors. This is because 1) the peptide release sites are not known, other than that they are not released into a synaptic cleft, 2) peptide access to receptors is unknowable from anatomy alone, 3) there might be additional, not yet identified hugin receptors, and 4) the presence of receptors alone does not prove a functional relationship because those receptors might be present to bind instead with different members of that peptide family released from other neurons. Concisely, anatomical results suggest but cannot prove functional consequences of peptidergic transmission. This latter point is in fact made explicitly/appropriately at several locations in the manuscript."

Another expert reviewer added, "The (putative) presence of both fast and slow synaptic transmission in these neurons isn't surprising, as it is something common to many neuropeptide neurons. Still, this is an important observation, and it should prompt a discussion about the role of neurotransmitters vs. peptide neuromodulators in feeding behavior. In the mammalian hypothalamus, GABA transmission is thought to be the main player in feeding regulation in the POMC and NPY neurons. Adding a few sentences on this would enrich the discussion. Importantly, the identity of the neurotransmitters involved in fast synaptic transmission isn't mentioned." (Is there any evidence that the small molecule co-transmitter is the same for each hugin neuron, both within and between modules?) "This seems a big omission given the scope of the article and that it would be straightforward experiment to do; I would encourage the authors to strengthen the manuscript by addressing these experiments at least in the context of a few of the Hugin microcircuits."

On this same point the first of the reviewers added, "The concept of co-transmission, a pivotal aspect of this paper as evident from its title, has been in the literature for several decades yet it does remain an under-developed area of study. Nevertheless, there are a number of published studies of the physiological influence of identified small molecule and neuropeptide co-transmitters on neural networks at the single cell level [none of which are referenced in this paper (?)]." Given the presence of this small but actual literature, laying out the context in which the hugin system can contribute would seem important, and its omission is unscholarly.

4) We would also like to see the authors address the relevance of the findings to adult feeding behavior. Assays for feeding behaviors in the larvae are still very primitive compared to those available in adults. Furthermore, larvae feeding is a fundamentally different process from adult feeling.

5) One expert reviewer stated "The description of the connectivity is only partial and the authors focus on the sensory inputs to the hugin-PC and hugin-VNC neurons. The description of the presynaptic circuitry of the other two hugin neuron types and of the postsynaptic circuitry of hugin-VNC neurons is incomplete. For example, hugin-RG and hugin-PH neurons have 39 and 23 presynaptic partners, respectively but the identities of there neurons are not described. Can the authors conclude something about what inputs these cells receive? Likewise, the postsynaptic partners of hugin-VNC neurons are not described in any detail. It seems that hugin-PC and hugin-VNC neurons also receive many input from non-sensory cells. What are these? The authors may of course choose what to show, but it should be stated clearly e.g., that the identity of these neurons is not known or that these circuits will be described elsewhere."

[Editors' note: further revisions were requested prior to acceptance, as described below.]

Thank you for resubmitting your work entitled "Synaptic transmission parallels neuromodulation in a central food-intake circuit" for further consideration at *eLife*. Your revised article has been favorably evaluated by Eve Marder (Senior editor), a Reviewing editor, and three reviewers.

The manuscript has been improved but there are some remaining issues that need to be addressed before acceptance, as outlined below. You will note that these mainly require editorial rewriting to reflect what appears still to be lack of consensus about what the essential message of the paper is, and still some tendencies to overstate what can be taken from the data. The Reviewing editor has summarized the results of the reviewer discussion, and also given you the entire reviews, for your information. We expect to make a final decision without returning the manuscript to the reviewers after receiving this next revision.

This is a very good revision of the previous submission that answers many of the concerns of the previous review, notably by adding new data implicating Hugin ACh co-transmission. Still there is some frustration that the authors have not precisely defined the focus of the paper, that they have pushed the homology with mammalian systems too far, that there are some irregularities in the RNAi data of Figure 3, and that the anatomical and genetic co-transmission data while strongly supportive are not backed up by cellular physiology studies performed in other systems. Quoting from the reviewer discussion:

1) '…the authors appear to be uncertain what is most important to emphasize (homology, cotransmission, connectomics, or feeding behavior) in their discussion/conclusions. I concur with the others that the homology argument is way oversold and the authors should be encouraged to back off considerably on that perspective; plus, doing so does not diminish the value of their work. The title of the paper is focused on cotransmission, which the authors do establish at the basic level (presence of cotransmitters, and their individual impact on feeding behavior when separately eliminated), but it seems to me that the heart of the paper is not about cotransmission or homology but about determining the input and output connectivity for the 4 classes of hugin-containing neurons. Should not the latter then be the focus of the paper's title?"

2) "As I argued in my first review, the authors cannot conclude that these circuits are homologous. The peptides (NMU/hugin) are orthologs (although one to many, since there is also a fly pyrokinin), the broad regions (SEZ, brain stem) are possibly homologous (but this is already highly contentious). But I don't think they can homologize a projection of a peptidergic cell to one area in the fly with cells expressing a homologous peptide in a (possibly) homologous brain region in the mouse. To claim circuit homology without even knowing the circuitry in a vertebrate is even more problematic. However, I think it is interesting to propose that there might be a common ancestry of these circuits, and there are clearly interesting parallels (including the neuropeptide targets of hugin/NMU). This could stimulate further work. However, it cannot be the main selling point of this paper."

3) "…the resolution of feeding experiments is unsatisfying, but I also recognize this is not an inherent fault of the paper because it is not its main goal. I do believe that the experiments added make it a better paper, although it seems that the RNAi data showed in that Figure 3 were quickly patched up for revision: some of it was previously published and the control is a mix of 3 different genetic backgrounds – this is highly unorthodox in the field and also I believe one should just replicate one's own experiment with new data instead of filling in old data, this is *Drosophila*, we are talking of a few crosses that take at best 2 weeks…."

The detailed reviews of the reviewers are appended and the authors should consider all the feedback provided in their revision.

Reviewer #1:

The revised manuscript is greatly improved. The authors added new data to show that ACh and hugin are both involved in the effects of hugin neurons on feeding. The authors also improved the discussion about the evolutionary significance of their findings.

This is an impressive body of work. The main problem is that there is a disconnect between the known behavioral effects of hugin neurons and the circuitry. The incomplete circuits shown in the paper do not explain how hugin influences feeding and locomotion.

We learn that there are four classes of hugin neurons that form distinct micro-circuits. However, it is not clear from these microcircuits, how hugin exerts its effects on feeding and locomotion.

1) Hugin-VNC neurons can increase locomotion motor rhythms. These neurons have many partners in the VNC and receive sensory input from abdominal and antennal nerves. However, it is not clear how they exert their effect on locomotion.

2) Hugin-PC neurons modulate feeding behavior and are necessary for the processing of bitter gustatory cues. For these neurons the EM reconstructions identified direct sensory inputs from putative gustatory neurons. This is a case where the EM analysis is revealing and suggests a direct sensory input from bitter sensory neurons to hugin-PC. Hugin-PC neurons project to the pars intercerebralis and synapse on neurosecretory cells. These connections are analyzed in detail in the paper. The pars intercerebralis cells express a hugin receptor and are activated by pharmacologically applied hugin. However, it is not clear if these connections are involved in the regulation of feeding or not. The authors show that hugin-PCs also have outputs in the SEZ (unknown interneurons), a center that houses the basic neuronal circuits generating feeding behavior. This suggests that the physiological role of the connection between hugin-PCs and the pars intercerebralis neurons is unknown. The authors should at least speculate about the role of these neurosecretory neurons in the context of feeding/locomotion. Alternatively, could hugin neurons have additional roles, e.g., in the regulation of metabolism or diuresis?

3) Hugin-RG neurons project to the ring gland and have neuroendocrine release sites bordering haemal space. They receive input from unidentified interneurons. It is not discussed what may be the role of secreted hugin.

4) For Hugin-PH neurons there were no targets identified as these are outside the sampled volume. The input neurons are also mostly unidentified. A discussion of the potential role of these neurons is lacking.

It took me quite a while to put this puzzle together, and the authors don't help the reader to understand what the circuitry can and cannot explain. I suggest that the authors provide a summary table with the four hugin cell types, their known effects, identified partners, putative effects as suggested by the EM analysis, and maybe the unknowns/suggestions for further experiments.

Reviewer #2:

This revision is a satisfying upgrade, particularly the identification of ACh as a small molecule cotransmitter for the hugin neurons. Many of the persisting issues that remain noteworthy relate to inappropriately equating anatomical results with physiological function. Some of these issues are readily repaired by modifying a conclusion from being firm (e.g. "indicates") to supportive (e.g. "suggests" or "supports the hypothesis that" or "is likely to", etc.). For example, the functional consequences of the co-localized ACh/Hugin are only likely or possible ones, not firm ones, because the appropriate experimental manipulations have yet to be performed. Similarly, other issues of this type would benefit from a slight change in the precision of the wording.

Regarding including appropriate literature citations for publications establishing neurotransmitter/neuropeptide co-transmission from identified neurons onto identified targets (see Author Reply to Reviewer Comments, Essential Revision #3, and the new Discussion), there is a strong selection from which to choose. I provide here an assortment (but not exhaustive list) of them, listed alphabetically, from several model systems published in well-regarded journals across the past 30 years. An even longer list of very nice publications can be generated regarding the functional consequences of this type of cotransmission from identified neurons at the level of network and/or behavioral output.

Blitz and Nusbaum, 1999, J Neurosci

Chalansani et al., 2010, Nat Neurosci

Ignell et al., 2009, PNAS [Flies]

Koh et al., 2003 J Neurophysiol

Li and van den Pol, 2006 J Neurosci

Qiu et al., 2016, *eLife*

Root et al., 2011, Cell [Root]

Sigvardt et al., 1986, J Neurosci

Stein et al., 2007, Eur J Neurosci

Sun et al., 2003, J Neurosci

Vilim et al., 1996, 2000, J Neurosci

Whim and Lloyd, 1989, PNAS

Results section paragraph two/Figure 2—figure supplement 1: How is it that there are "presynaptic densities" associated with clusters of dense cored vesicles (DCVs) and not small clear vesicles (SCVs) in the hugin-RG neuron terminals in the ring gland? Both in this manuscript and throughout the literature, it is established that presynaptic densities in neurons are the domain of SCVs and their release, and that DCVs are neither clustered close to the plasma membrane (or elsewhere) nor is there any ultrastructural identifier of their sites of release. Is my understanding of the vesicle cluster in that figure not accurate? If accurate, is there precedence for this relationship in the literature?

Subsection “Hugin classes form distinct units that share synaptic partners” and “Organizational principles of a peptidergic network” and elsewhere: Axo-axonic connections. Are there truly axo-axonic connections (as this type of connection is classical defined in the literature) onto hugin neurons? This type of synapse commonly refers to synaptic sites located close to specific transmitter release sites at axon terminals/boutons, where the axo-axonic synapse selectively influences only that transmitter release site and possibly nearby sites. These types of synapses are also electrotonically distant/isolated from the population of synaptic inputs which collectively determine whether or not the target neuron will fire an action potential, so they are not part of the synaptic integration mechanism that determines neuronal activity. There appears to be no explicit segregation in the text regarding axonal membrane vs. "synaptic integration membrane" or what would be called dendritic membrane in a vertebrate. However, as is common in invertebrate nervous systems, the hugin neurons do have neuropilar membrane on which there is an intermingling of inputs and outputs, as indicated in the second paragraph of the Results section.

Results section: "Hugin-PC and hugin-VNC neurons' projections represent mixed synaptic input-output compartments as they both showed pre- as well as postsynaptic sites along their neurites (Figure 2)."

If there are truly axo-axonic synapses onto hugin neurons, then please provide more explicit evidence.

Overall, the core results of this paper well-establishes the (ACh/)hugin connectome and provides an interesting parallel to the distribution and behavioral influences of the mammalian hugin-equivalent (neuromedin U; NMU) peptide-containing neurons, which should be of interest to the broad *eLife* readership.

Reviewer #3:

The authors' additions strengthened the manuscript and provided tools and avenues of investigation for other scientists in the field.

1) Co-transmission

The ChAT experiments, especially the genetic data are quite interesting. While it is still far from clear how the two modes of transmission work and how they work together to mediate the Hugin neurons effect on behavior, I recognize this is outside the scope of this manuscript. It seems that the hugin neurons may be a good model to study this. The new parts of the text in the main body and discussion that address this were satisfactory.

---

## [Author Response]

*[…]*

*Essential revisions:*

*There are some reservations about this paper that we ask the authors to address in revision.*

*1) The 4 hugin microcircuits likely have different functions, but are there any indications of what these might be? Can the authors provide any evidence that might begin to identify these functions? Is there evidence supporting a feeding-related function for each of the 4 hugin modules? Commonly, neuropeptides (like other transmitter molecules) subserve neural signaling in numerous unrelated contexts.*

We understand the relevance of this issue. In fact, there already is a body of evidence demonstrating that each hugin module has its own function and that not all of them are related to feeding. Just recently Hückesfeld et al., 2016 showed that only hugin-PC neurons are involved in processing of bitter gustatory cues and subsequent reduction of food intake. Previously, Schoofs et al., 2014 showed that hugin-VNC neurons are solely responsible for increase in locomotion motor pattern. These data fit to our finding of independent microcircuits for each hugin class. We have added references to above publications at relevant points throughout the manuscript and hope that this evidence is sufficient to address the reviewers’ concern.

*2) There is concern that the homology/analogy with the mammalian NMU neuron is overemphasized. Quoting from one of the expert reviews, "The authors compare the circuitry of hugin neurons to the connectivity of NMU neurons in vertebrates. Homologizing neuron types and brain regions across large evolutionary distances is difficult. The authors are apparently aware of these difficulties and sometimes use the term homology and sometimes analogy. These are of course not the same.*

We believe our usage of ‘homologous’ and ‘analogous’ to be in agreement with current literature. However, we are aware of the difficulties when comparing across large evolutionary distances and have adjusted respective paragraphs and figure legends to better reflect that. In addition, the Introduction was adjusted to better convey the difference between analogous and homologous (Figure 1). We hope that our work may help in sparking a positive and constructive discussion on this matter.

*One criterion for establishing homologous neuron types across phyla would be that the neurons express a similar set of transcription factors and effector genes and are located in brain areas that as regions are possibly homologous (e.g., the spinal cord and the VNC).*

*The difficulty in comparing the fly hugin neurons to the NMU neurons in vertebrates is that in vertebrates the expression pattern of NMU is more complex. NMU neurons can be found in the hypothalamus, the pituitary, the brainstem, the spinal cord, and throughout the gastrointestinal tract. However, in the fly, hugin+ neurons are only found in the SEZ. These project to the VNC, PC and RG, but there are no hugin+ somata in these regions. The authors compare the projections of the hugin neurons to the distinct sets of NMU+ neurons in vertebrates. This is problematic. It is only the hugin+ cells in the SEZ in Drosophila that may be comparable to NMU+ cells in the brainstem in vertebrates. In this sense, Figure 1 is misleading. The authors must be more careful about this and explain more carefully what they are trying to compare to what and based on which criteria. Regarding a comparison of circuitry, it is even more problematic, since not much is known about the circuitry of brainstem NMU neurons in vertebrates.*

*The similarity in chemical connectivity of hugin/NMU-DH44/CRF neurons between the fly and vertebrates is intriguing. However, it is again difficult to directly compare these circuits, since there are no hugin+ neurons in the pars intercerebralis that would be equivalent to the hypothalamic NMU neurons."*

We agree that the NMU system in mammals is obviously far more complex than hugin in *Drosophila* and that this difference has to be pointed out more clearly. We have adjusted the Introduction and Figure 1 to reflect the fact that we compare occurrence of NMU (transcript or peptide) with projection targets of the different types of hugin neurons. We also added an additional paragraph to the Discussion section pointing out differences between NMU and hugin.

*3) Co-transmission as a theme is only partially supported and weakly contextualized. Quoting from one of the expert reviewers, "The conclusions regarding the functional consequences of the co-localized (not yet identified) small molecule transmitter and hugin peptide are only likely or possible ones, not firm ones. For example, the extent to which the influence of the hugin neurons, and particularly that of any one of the 4 modules, mimics the previously documented behavioral consequences of genetically manipulating the presence of the hugin peptide remains to be determined. Also, the proposed target neurons of neuronally released hugin peptide are the likely but not necessarily actual target neurons, despite the presynaptic presence of hugin peptide and postsynaptic presence of hugin receptors. This is because 1) the peptide release sites are not known, other than that they are not released into a synaptic cleft, 2) peptide access to receptors is unknowable from anatomy alone, 3) there might be additional, not yet identified hugin receptors, and 4) the presence of receptors alone does not prove a functional relationship because those receptors might be present to bind instead with different members of that peptide family released from other neurons. Concisely, anatomical results suggest but cannot prove functional consequences of peptidergic transmission. This latter point is in fact made explicitly/appropriately at several locations in the manuscript."*

*Another expert reviewer added, "The (putative) presence of both fast and slow synaptic transmission in these neurons isn't surprising, as it is something common to many neuropeptide neurons. Still, this is an important observation, and it should prompt a discussion about the role of neurotransmitters vs. peptide neuromodulators in feeding behavior. In the mammalian hypothalamus, GABA transmission is thought to be the main player in feeding regulation in the POMC and NPY neurons. Adding a few sentences on this would enrich the discussion. Importantly, the identity of the neurotransmitters involved in fast synaptic transmission isn't mentioned." (Is there any evidence that the small molecule co-transmitter is the same for each hugin neuron, both within and between modules?) "This seems a big omission given the scope of the article and that it would be straightforward experiment to do; I would encourage the authors to strengthen the manuscript by addressing these experiments at least in the context of a few of the Hugin microcircuits."*

*On this same point the first of the reviewers added, "The concept of co-transmission, a pivotal aspect of this paper as evident from its title, has been in the literature for several decades yet it does remain an under-developed area of study. Nevertheless, there are a number of published studies of the physiological influence of identified small molecule and neuropeptide co-transmitters on neural networks at the single cell level [none of which are referenced in this paper (?)]." Given the presence of this small but actual literature, laying out the context in which the hugin system can contribute would seem important, and its omission is unscholarly.*

We tested one of the most abundantly expressed small molecule neurotransmitter in the fly brain, acetylcholine (ACh), and found that subsets of hugin neurons are cholinergic. Knockdown of ACh using RNAi rescued the hugin phenotype just as a knockdown of the hugin neuropeptide (the latter having already been demonstrated by Schoofs et al., 2014). This shows the presence and function of both synaptic transmitter and peptidergic transmission. The data has been compiled into a new figure (Figure 3) and a new section of the Results (subsection “Acetylcholine is a co-transmitter in hugin neurons”). We believe that these findings on ACh in hugin neurons strengthen the concept of co-transmission.

To also consolidate putative targets of the hugin neuropeptide, we conducted pharmacological experiments with synthetic hugin peptide and showed that calcium activity in the neurosecretory cells is indeed increased. This data has been compiled into a new figure (Figure 8—figure supplement 2). Nevertheless, as this reviewer pointed out, we are fully aware of the limitations of our data regarding the question of where exactly the hugin neuropeptide is being released.

Regarding literature on co-transmission: we understand that by focusing on studies that have looked at this issue at a similar level of resolution, we have omitted prominent cases of co-transmission from the mammalian field. We adjusted the discussion to better represent current examples of co-transmission throughout different systems by adding further references (Cansell et al., 2015; Telegdy and Adamik 2013; Tanaka and Telegdy 2014). Should we still be missing important references, we would appreciate further suggestions on which references to include.

*4) We would also like to see the authors address the relevance of the findings to adult feeding behavior. Assays for feeding behaviors in the larvae are still very primitive compared to those available in adults. Furthermore, larvae feeding is a fundamentally different process from adult feeling.*

We have added a new paragraph to the Discussion to address this issue. However, we feel compelled to point out that, on the contrary, assays in larvae – specifically the ones employed in the study of hugin – are at least on par with those in adult flies.

*5) One expert reviewer stated "The description of the connectivity is only partial and the authors focus on the sensory inputs to the hugin-PC and hugin-VNC neurons. The description of the presynaptic circuitry of the other two hugin neuron types and of the postsynaptic circuitry of hugin-VNC neurons is incomplete. For example, hugin-RG and hugin-PH neurons have 39 and 23 presynaptic partners, respectively but the identities of there neurons are not described. Can the authors conclude something about what inputs these cells receive? Likewise, the postsynaptic partners of hugin-VNC neurons are not described in any detail. It seems that hugin-PC and hugin-VNC neurons also receive many input from non-sensory cells. What are these? The authors may of course choose what to show, but it should be stated clearly e.g., that the identity of these neurons is not known or that these circuits will be described elsewhere."*

It is true that a large fraction of synaptic partners of hugin neurons are interneurons. At this point, we are unable to draw any conclusion just based on their morphology. Future investigations may be able to shed light on these circuits. By providing a detailed neuron atlas, we hope that other labs will be able to pick up on our work. For now we have added more details on the fraction of interneurons and made it clear that – at this point – it is difficult to draw conclusions.

*[Editors' note: further revisions were requested prior to acceptance, as described below.]*

*[…]*

*This is a very good revision of the previous submission that answers many of the concerns of the previous review, notably by adding new data implicating Hugin ACh co-transmission. Still there is some frustration that the authors have not precisely defined the focus of the paper, that they have pushed the homology with mammalian systems too far, that there are some irregularities in the RNAi data of Figure 3, and that the anatomical and genetic co-transmission data while strongly supportive are not backed up by cellular physiology studies performed in other systems. Quoting from the reviewer discussion:*

*1) '…the authors appear to be uncertain what is most important to emphasize (homology, cotransmission, connectomics, or feeding behavior) in their discussion/conclusions. I concur with the others that the homology argument is way oversold and the authors should be encouraged to back off considerably on that perspective; plus, doing so does not diminish the value of their work. The title of the paper is focused on cotransmission, which the authors do establish at the basic level (presence of cotransmitters, and their individual impact on feeding behavior when separately eliminated), but it seems to me that the heart of the paper is not about cotransmission or homology but about determining the input and output connectivity for the 4 classes of hugin-containing neurons. Should not the latter then be the focus of the paper's title?"*

The homology aspect has been strongly deemphasized throughout the manuscript (specifically in respect to the networks’ architecture). We agree that the manuscript touches a broad range of issues. However, we consider the hugin connectome to be similar to e.g. a genetic screen which, while being at the heart of the paper, does not represent the most interesting finding. Instead we decided to further pursue one striking aspect of the hugin connectome by investigating the co-transmission along the hugin-endocrine axis.

*2) "As I argued in my first review, the authors cannot conclude that these circuits are homologous. The peptides (NMU/hugin) are orthologs (although one to many, since there is also a fly pyrokinin), the broad regions (SEZ, brain stem) are possibly homologous (but this is already highly contentious). But I don't think they can homologize a projection of a peptidergic cell to one area in the fly with cells expressing a homologous peptide in a (possibly) homologous brain region in the mouse. To claim circuit homology without even knowing the circuitry in a vertebrate is even more problematic. However, I think it is interesting to propose that there might be a common ancestry of these circuits, and there are clearly interesting parallels (including the neuropeptide targets of hugin/NMU). This could stimulate further work. However, it cannot be the main selling point of this paper."*

We fully agree with the reviewer’s point of view. We believed that the last revision of the paper had been sufficient to prevent over-interpretation of our suggestions. As stated above, we have now further deemphasized the homology aspect and speculations about potential similarities of these circuits have been rephrased to clarify that existing data on vertebrate NMU circuits is insufficient to make definite statements.

*3) "…the resolution of feeding experiments is unsatisfying, but I also recognize this is not an inherent fault of the paper because it is not its main goal. I do believe that the experiments added make it a better paper, although it seems that the RNAi data showed in that Figure 3 were quickly patched up for revision: some of it was previously published and the control is a mix of 3 different genetic backgrounds – this is highly unorthodox in the field and also I believe one should just replicate one's own experiment with new data instead of filling in old data, this is Drosophila, we are talking of a few crosses that take at best 2 weeks…."*

We apologize for any confusion the explanation of these experiments may have caused. The RNAi data were in fact not quickly patched up for revision. Instead they were obtained as part of a larger genetic screen which was performed in 2014 and a part of that screen was published in Schoofs et al., 2014. Back then, the ChAT RNAi data was not included because our knowledge of the hugin neurons was insufficient to explain these results. Sticking to the original data including controls performed at that time, was a deliberate decision to preserve consistency. There was no filling in of old data. We have rephrased unclear passages and added more information on this to Materials and methods.

*The detailed reviews of the reviewers are appended and the authors should consider all the feedback provided in their revision.*

*Reviewer #1:*

*[…]*

*It took me quite a while to put this puzzle together, and the authors don't help the reader to understand what the circuitry can and cannot explain. I suggest that the authors provide a summary table with the four hugin cell types, their known effects, identified partners, putative effects as suggested by the EM analysis, and maybe the unknowns/suggestions for further experiments.*

We added a new paragraph and Table 1 to the Discussion which summarize and assess connectivity vs. function for each hugin.

*Reviewer #2:*

*This revision is a satisfying upgrade, particularly the identification of ACh as a small molecule cotransmitter for the hugin neurons. Many of the persisting issues that remain noteworthy relate to inappropriately equating anatomical results with physiological function. Some of these issues are readily repaired by modifying a conclusion from being firm (e.g. "indicates") to supportive (e.g. "suggests" or "supports the hypothesis that" or "is likely to", etc.). For example, the functional consequences of the co-localized ACh/Hugin are only likely or possible ones, not firm ones, because the appropriate experimental manipulations have yet to be performed. Similarly, other issues of this type would benefit from a slight change in the precision of the wording.*

*Regarding including appropriate literature citations for publications establishing neurotransmitter/neuropeptide co-transmission from identified neurons onto identified targets (see Author Reply to Reviewer Comments, Essential Revision #3, and the new Discussion), there is a strong selection from which to choose. I provide here an assortment (but not exhaustive list) of them, listed alphabetically, from several model systems published in well-regarded journals across the past 30 years. An even longer list of very nice publications can be generated regarding the functional consequences of this type of cotransmission from identified neurons at the level of network and/or behavioral output.*

*Blitz and Nusbaum, 1999, J Neurosci*

*Chalansani et al. (2010, Nat Neurosci*

*Ignell et al., 2009, PNAS [Flies]*

*Koh et al., 2003, J Neurophysiol*

*Li and van den Pol, 2006, J Neurosci*

*Qiu et al., 2016, eLife*

*Root et al., 2011, Cell [Root]*

*Sigvardt et al., 1986, J Neurosci*

*Stein et al., 2007, Eur J Neurosci*

*Sun et al., 2003, J Neurosci*

*Vilim et al., 1996, 2000) J Neurosci*

*Whim and Lloyd, 1989, PNAS*

We would like to thank the reviewer for this extensive list of suggested references. After careful consideration, we have added Stein et al., 2007; Sun et al., 2003; Whim and Lloyd, 1989; Li and van den Pol, 2006 and Koh et al., 2003.

*Results section paragraph two/Figure 2—figure supplement 1: How is it that there are "presynaptic densities" associated with clusters of dense cored vesicles (DCVs) and not small clear vesicles (SCVs) in the hugin-RG neuron terminals in the ring gland? Both in this manuscript and throughout the literature, it is established that presynaptic densities in neurons are the domain of SCVs and their release, and that DCVs are neither clustered close to the plasma membrane (or elsewhere) nor is there any ultrastructural identifier of their sites of release. Is my understanding of the vesicle cluster in that figure not accurate? If accurate, is there precedence for this relationship in the literature?*

We agree that this phrasing is misleading and indeed we do not know what exactly these membrane specializations are. We have thus rephrased ‘presynaptic densities’ to ‘membrane specializations resembling presynaptic densities’ in text body as well as the legend of Figure 2—figure supplement 1.

*Subsection “Hugin classes form distinct units that share synaptic partners” and “Organizational principles of a peptidergic network” and elsewhere: Axo-axonic connections. Are there truly axo-axonic connections (as this type of connection is classical defined in the literature) onto hugin neurons? This type of synapse commonly refers to synaptic sites located close to specific transmitter release sites at axon terminals/boutons, where the axo-axonic synapse selectively influences only that transmitter release site and possibly nearby sites. These types of synapses are also electrotonically distant/isolated from the population of synaptic inputs which collectively determine whether or not the target neuron will fire an action potential, so they are not part of the synaptic integration mechanism that determines neuronal activity. There appears to be no explicit segregation in the text regarding axonal membrane vs. "synaptic integration membrane" or what would be called dendritic membrane in a vertebrate. However, as is common in invertebrate nervous systems, the hugin neurons do have neuropilar membrane on which there is an intermingling of inputs and outputs, as indicated in the second paragraph of the Results section.*

*Results section: "Hugin-PC and hugin-VNC neurons' projections represent mixed synaptic input-output compartments as they both showed pre- as well as postsynaptic sites along their neurites (Figure 2)."*

*If there are truly axo-axonic synapses onto hugin neurons, then please provide more explicit evidence.*

Synapses between hugin interneurons (PC/VNC) are found along their main neurites as are other presynaptic sites. In this, it is according to the situation described by the reviewer. However, these hugin neurons are very unpolar and do not have clearly defined axonal/dendritic compartments. We therefore agree that the use of the term axo-axonic is problematic and we have rephrased this accordingly throughout the manuscript.